# Enhancing Contrastive Learning for Ordinal Regression via Ordinal Content Preserved Data Augmentation

**Jiyang Zheng**[1,2]    **Yu Yao**[3,4]    **Bo Han**[5]    **Dadong Wang**[2]    **Tongliang Liu**[1]*

[1]Sydney AI Center, The University of Sydney [2]CSIRO, Data61
[3]Mohamed bin Zayed University of Artificial Intelligence
[4]Carnegie Mellon University [5]Hong Kong Baptist University
`jzhe5740@uni.sydney.edu.au  yu.yao@mbzuai.ac.ae`
`dadong.wang@data61.csiro.au  tongliang.liu@sydney.edu.au`

## Abstract

Contrastive learning, while highly effective for a lot of tasks, shows limited improvement in ordinal regression. We find that the limitation comes from the predefined strong data augmentations employed in contrastive learning. Intuitively, for ordinal regression datasets, the discriminative information (ordinal content information) contained in instances is subtle. Existing strong augmentations can easily overshadow this ordinal content information. As a result, when contrastive learning is used to extract common features between weakly and strongly augmented images, the derived features often lack this essential ordinal content, rendering them less useful in training models for ordinal regression. To improve contrastive learning's utility for ordinal regression, we propose a novel augmentation method to replace the predefined strong argumentation based on the principle of *minimal change*. Our method is designed in a generative manner that can effectively generate images with different styles but contains desired ordinal content information. Extensive experiments validate the effectiveness of our proposed method, which serves as a plug-and-play solution and consistently improves the performance of existing state-of-the-art methods for ordinal regression.

## 1 Introduction

Contrastive learning encourages feature consistency between strongly and weakly augmented versions of data (Chen et al., 2020a). This method has demonstrated its effectiveness in both supervised and unsupervised learning contexts (Oord et al., 2018; He et al., 2019; Khosla et al., 2020).

However, the efficacy of contrastive learning diminishes when applied to ordinal regression, which aims to predict ordered categories from instances. A primary reason for this limited success is that contrastive learning does not fully account for the unique characteristics of ordinal regression data. Unlike conventional classification data, instances in ordinal regression datasets are inherently characterized by their subtle ordinal content information, which is essential for predicting ordinal labels. This subtlety presents a significant challenge for contrastive learning to effectively learn discriminative features, primarily due to the predefined strong data augmentations such as color jittering and color dropping (Chen et al., 2020a; Von Kügelgen et al., 2021; Xiao et al., 2021). These augmentations can easily overshadow or diminish the vital ordinal content information.

Take an ordinal regression task of age estimation as an example. While images used in this task may be rich in pixels and colors, age-discriminate features, such as wrinkles or gray hair, are often localized to small regions. This localization renders the age-related content information subtle, making it susceptible to being overshadowed by strong augmentations. As shown in Figure 1, the commonly used strong augmentations can distort or even erase these essential features in ordinal regression data, compromising the performance of contrastive learning in such tasks. Similarly, in

---

*Corresponding Author

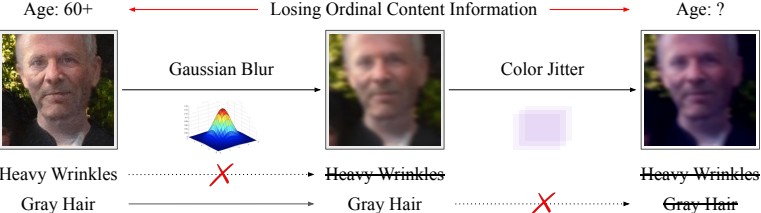

Figure 1: Ordinal content information in data can be easily distorted by standard augmentations in contrastive learning. As illustrated in the example, the age-related features: wrinkles and hair color are eliminated after Gaussian blurring and color jitter, making the age become unidentifiable.

the task of diabetic retinopathy grading, the parts in the retina images that indicate the severity of the condition, such as microaneurysms or hemorrhages (Salz et al., 2016), are often localized and subtle, which can be simply removed by augmentation operations such as Gaussian blurring.

To enhance contrastive learning for ordinal regression, we propose a novel augmentation method as an alternative to predefined strong augmentations. Our method is in a generative manner which can effectively preserve ordinal content information. After training with ordinal datasets, it can be leveraged to generate new instances with the same ordinal content information but different styles. To achieve this, we employ a generative model combined with the principle of *minimal change*.

Specifically, a Generative Adversarial Network (GAN) is employed to generate new instances. However, this generative model does not naturally preserve ordinal content information during the generation process. As indicated by existing work (Hyvarinen et al., 2019; Khemakhem et al., 2020; Von Kügelgen et al., 2021), without any constraints, learned latent factors entangle both ordinal content and non-ordinal information. Such entanglement compromises the controllability of the generative process, making it challenging to produce instances with specific ordinal information. To address this, we introduce the *minimal change* principle (Xie et al., 2022; Kong et al., 2022), which claims that *changes to latent factors should have minimal impact on the generation of instances*.

By leveraging the minimal change, we effectively separate ordinal content factors from non-ordinal ones. In our approach, we partition the latent factors into two sets: $\hat{z}_o$ and $\hat{z}_n$. The former, $\hat{z}_o$, serves as a "container" for ordinal content information. The latent factors in this set are essential for determining the ordinal category and are generated in alignment with ordinal labels. Minimal change is applied to $\hat{z}_o$, thereby limiting its influence during the generation of new instances. As a result, once the generative model attains a minimal reconstruction error, $\hat{z}_o$ solely captures essential ordinal content information, while $\hat{z}_n$ encompasses the remaining non-ordinal information (information that does not contribute to determining ordinal labels). This results in the successful disentanglement of ordinal and non-ordinal information. A more detailed explanation of this disentanglement, along with the generative process introduced by our method, is provided in Section 3.

Our method enhances existing ordinal regression methods by harnessing the benefits of contrastive learning through ordinal content-preserving augmentations. After learning the latent factors $\hat{z}_o$ and $\hat{z}_n$ with the minimal change principle, our method creates augmented instances that belong to a specific ordinal label. To achieve it, we fix $\hat{z}_o$ and randomly sample $\hat{z}_n$, feeding them into our trained generator. This process produces instances with varied styles but consistent ordinal information. These instances can serve as strong augmented views to the natural images. By utilizing these instances with a contrastive learning loss, our method can be seamlessly integrated into existing ordinal regression methods.

The remaining paper is structured as follows: Section 2 provides an overview of the related works in ordinal regression and contrastive learning. Section 3 details our proposed methodology and its practical implementations. Section 4 includes the experimental validation of our approach on different ordinal regression tasks. Section 5 concludes our paper.

## 2    RELATED WORKS

**Method for Ordinal Regression.**    Recent advancements often frame ordinal regression as a classification task (Niu et al., 2016; Liu et al., 2017; Beckham & Pal, 2017). Liu et al. (2018b) introduced

a constrained optimization formulation for ordinal regression. This approach minimizes the negative log-likelihood across multiple classes while simultaneously preserving the inherent order relationship between instances. Diaz & Marathe (2019) leveraged the natural ordinal relationships between targets, imparting them as prior knowledge to the model through soft labels. Liu et al. (2019) addressed the task from a probabilistic modeling perspective, where a Gaussian Process model is attached to the output layer of the deep neural network to model uncertainty. Li et al. (2021) proposed a framework that employs probabilistic embeddings to model data uncertainty. Their method enforces a constraint between the learned embedding distributions and pre-defined ordinal distributions, ensuring that the learned embedding space remains ordered. Shin et al. (2022) proposed a regression-based rank estimation algorithm that learns to model the order relationship between instances. Cheng et al. (2023) propose a data fusion approach to address the class-imbalance issue in ordinal regression datasets. While most of the previous studies primarily focused on aligning the model's final predictions with the target, our approach emphasizes the importance of preserving ordinal content information when augmenting ordinal regression data. Our method is orthogonal to end-to-end trainable ordinal regression model, which can serve as a plug-and-play solution to improve the performance of existing state-of-the-art ordinal regression frameworks.

**Contrastive Learning and Data Augmentations.** Contrastive learning (CL) extracts discriminative information from data by organizing samples into similar and dissimilar pairs. It amplifies the similarity of similar pairs and increases the difference between dissimilar pairs in the feature space. In a self-supervised setting (He et al., 2019; Chen et al., 2020a), similar pairs are generated through a data augmentation module. Given a reference image, this module introduces modifications such as random scaling, cropping, color jittering, blurring, and flipping to generate new perspectives of the image. The original image and its augmented versions constitute a similar pair. In contrast, other images in the batch are treated as dissimilar samples, forming dissimilar pairs. Khosla et al. (2020) expanded CL to a supervised setting. In addition to the augmented versions of the image, samples from the same class also become part of a similar pair. Zha et al. (2022) introduced a supervised CL framework for regression tasks, ensuring that the order of representations in the feature space corresponds to their target values. The results demonstrate that existing regression methods consistently benefit from a CL module for extracting discriminative features from data.

Furthermore, intensive data augmentations have been found crucial for the success of the contrastive learning framework across all settings (Chen et al., 2020a;b; Khosla et al., 2020; Li et al., 2023; 2022a; Huang et al., 2021; Zheng et al., 2022). However, such aggressive augmentations can compromise an image's content. Xiao et al. (2021) addressed this breach of the invariance assumption (i.e., data augmentations altering the data's semantic information) by decomposing a compound series of augmentations into individual operations and creating distinct heads for each single augmentation. This method is effective when data is sensitive to a few specific augmentations within the full augmentation sequence but maintains its invariance assumption with others. Given that content information in ordinal data is particularly susceptible, many augmentation techniques can potentially distort an image's semantics, thereby limiting the method's efficacy. Compared to their approach, our method does not depend on any predetermined augmentation method. Instead, we guide the model to discern which aspects of the augmented data should be preserved as content variables and which can be modified to introduce diversity as style variables.

## 3    Ordinal Content Preserving Contrastive Learning (OCP-CL)

In this section, we introduce our Ordinal Content Preserving Contrastive Learning (OCP-CL) framework. Specifically, to improve the utility of contrastive learning for ordinal regression, we propose a novel ordinal content-preserving augmentation method that replaces the predefined strong augmentations in a contrastive learning framework. First, we explain the generative process of ordinal regression data, which serves as the foundational understanding of our proposed generative model. Then, we introduce our approach for disentangling ordinal content and non-ordinal factors via minimal change, and detail the implementation of the generative model. Next, we describe the process of generating content-preserving data augmentations through interventions on non-ordinal latent factors. Finally, we present the contrastive learning formulation with our generated augmentations from the original instances. The contrastive learning objective can be integrated into any existing end-to-end trainable deep ordinal regression methods to form a joint objective.

**Data Generative Process.** We first explain the causal data generative process (Glymour & Zhang, 2019; Yao et al., 2023) as illustrated in Figure 2. The graph outlines the generative process for ordinal regression data, segregating latent factors into different functional groups based on their relationships to the observed variables.

Specifically, $z_v$ denotes a set of invariant ordinal factors that capture all ordinal content relevant features across different ordinal categories. For example, in age estimation, $z_v$ encompasses a comprehensive collection of age-specific attributes such as the severity of wrinkles or variations in skin texture across age groups. The ordinal label $y$ serves as a constraining variable, ensuring that $z_o$ selectively retains only those features from $z_v$ that pertain to its corresponding ordinal category. Similarly, $z_n$ denotes the set of non-ordinal, style-related factors. Together, $z_o$ and $z_n$ collaboratively generate $x$, the observed data instance. Our primary objective for the generative model is to disentangle the learned latent factors $\hat{z}$ into $\hat{z}_o$ and $\hat{z}_n$, in alignment with the proposed SCM. In this setup, $\hat{z}_o$ is approximated to only contain ordinal content information that determines the ordinal category, whereas $\hat{z}_n$ holds styling information. Generating content-preserving samples hinges on accurately recovering the true joint distribution of image and class, denoted as $P(X, Y)$. Based on our proposed generative process, the causal factorization of the joint distribution $P(X, Y, Z_o, Z_v, Z_n)$ is:

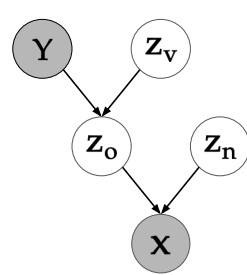

Figure 2: The data generative process employed by our method. The shaded variables are observable and the unshaded variables are latent.

$$P(X, Y, Z_o, Z_v, Z_n) = P(Y)P(Z_v)P(Z_n)P(Z_o|Y, Z_v)P(X|Z_o, Z_n). \tag{1}$$

Our method is designed to fulfill the this generative process by modelling each probability in Eq. 1.

### 3.1 Philosophy of Disentangling Ordinal Content Factors.

The principle of minimal change is pivotal in our generative process, aiding in the effective disentanglement of non-ordinal factors from ordinal content factors. To underscore the importance of this principle, we first discuss the limitations of relying solely on ordinal labels $y$. Subsequently, we explore scenarios that incorporate the *minimal change* principle. We consider a generative model (Zhou et al., 2023; Yao et al., 2021) that aligns with the generative process depicted in Figure 2 and infers the factors $\hat{z}_n$, $\hat{z}_o$, and $\hat{z}_v$. After the learning phase, in which the reconstruction error is minimized, our goal is align $\hat{z}_n$ with non-ordinal factors $z_n$; align $\hat{z}_o$ with ordinal content factors $z_o$; and align $\hat{z}_v$ with the set of invariant ordinal factors $z_v$. Note that $z_n$, $z_o$ and $z_v$ are the true latent factors in the data generative process.

If we rely solely on ordinal labels $y$, the generation of $\hat{z}_o$ is influenced by both $y$ and $\hat{z}_v$. This relationship can be mathematically expressed as: $\hat{z}_o = g(y, \hat{z}_v) + \epsilon$. In essence, this generation mechanism allows $\hat{z}_o$ to assimilate information from both $y$ and $\hat{z}_v$. Given the generative model's typical assumption that both $\hat{z}_n$ and $\hat{z}_v$ follow a standard Gaussian distribution, non-ordinal information can feasibly reside in either $\hat{z}_v$ or $\hat{z}_n$. These factors are fundamentally similar, and their differences are purely notational. Hence, the choice of where to store non-ordinal information does not influence the reconstruction error. As a result, $\hat{z}_o$, being influenced by both $y$ and $\hat{z}_v$, inevitably contains non-ordinal information. Thus, the disentanglement can not be achieved solely with $y$.

To address this, we apply the minimal change principle in the disentanglement process. The minimal change principle serves as a constraint on image generation, ensuring that the influence of certain factors during instance generation remains minimal. Specifically, when applied to $\hat{z}_o$, this principle limits the factor's influence. Consequently, the information within $\hat{z}_o$ remains minimal and focused. By introducing an additional constraint on the generative function $g$, we ensure that $\hat{z}_v$ inherently carries information related to $Y$. Assuming the reconstruction error is minimized, this suggests that ordinal content information is primarily included within the latent factors $\hat{z}_o$. As $\hat{z}_n$ is generated independently of the ordinal label $Y$, it will not contain ordinal content information. Therefore, all ordinal content information becomes localized within $\hat{z}_o$. Furthermore, by minimizing the influence of $\hat{z}_o$, it becomes exclusively representative of ordinal content information.

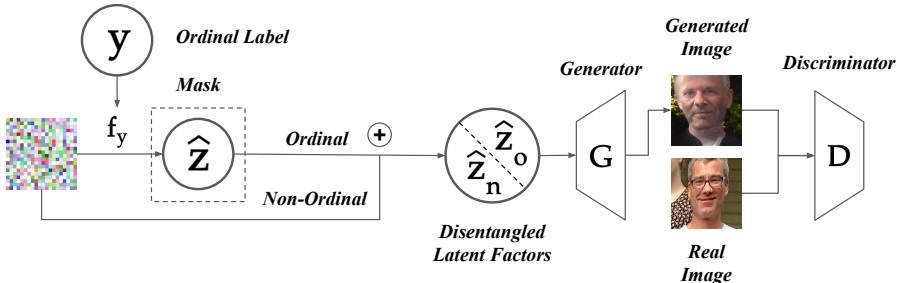

Figure 3: **Architecture of Our Generative Model.** The class label $y$ is leveraged for both disentangling latent factors and enforcing minimal change. An ordinal head is appended to the discriminator to preserve the ordinal distribution of generated samples in relation to their class.

## 3.2 ORDINAL CONTENT AND NON-ORDINAL INFORMATION DISENTANGLEMENT VIA MINIMAL CHANGE

Our primary objective is to achieve *minimal change* in the generation process, specifically by constraining the influence of ordinal content factors $\hat{z}_o$. This is realized by limiting the number of these factors. To this end, we introduce a mask operation, consistent with our data generative process. The essence of this mask is to regulate the quantity of ordinal content factors. A sparser mask translates to fewer ordinal content factors. To promote this sparsity, we impose an L1 loss on the mask, represented as $\mathcal{L}_{sp} = \|M\|_1$. Let's define our latent factors as $\hat{z} := [\hat{z}_o, \hat{z}_n]$ and $\hat{z}_{on} := [\hat{z}_v, \hat{z}_n]$. The mask operation is then given by:

$$\hat{z}_{on} = \hat{z} + M \odot f_y(\hat{z}), \ \ \hat{z} \sim \mathcal{N}(0, \mathbf{I}), \ \ y \sim P(Y). \tag{2}$$

In the above equation, $\hat{z}$ encapsulates both the invariant ordinal content factors $\hat{z}_v$ and the non-ordinal factors $\hat{z}_n$. The label distribution $P(Y)$ is derived empirically by counting the occurrences of each label within the dataset and then normalizing these counts to form a probability distribution. We utilize the Deep Sigmoidal Flow (Huang et al., 2018) for the function $f_y$. It is a type of normalization flow characterized by the use of small neural networks with sigmoid units. These units introduce inflection points in the transformation function, enabling the modeling of complex probability distributions. Within our context, it serves as a component-wise transformation function that transforms $\hat{z}_{on}$ in a component-wise manner. This function, when applied, acts as a label influence. Although it impacts all elements in $\hat{z}$, each element undergoes an independent transformation. The mask operation on $\hat{z}$ rejects certain transformed elements. By adding $\hat{z}$ back, we ensure that certain elements remain uninfluenced by the label $Y$, effectively distinguishing them as non-ordinal factors.

It's worth noting that for elements unaffected by the mask but influenced by $y$, the addition of elements from $\hat{z}$ is inconsequential. Given that $\hat{z}$ is sampled from a high-dimensional Gaussian distribution, adding it to these elements is akin to introducing random Gaussian noise, ensuring our method remains consistent with the proposed data generative process.

To make latent factors, we employ a Generative Adversarial Network (GAN) model (Mirza & Osindero, 2014). The architecture of this model is illustrated in Figure 3. The GAN comprises two main components: a generator $G_\theta$ and a discriminator $D_\phi$, each parameterized by their respective learnable parameters $\theta$ and $\phi$. The generator's role is to craft realistic instances, while the discriminator endeavors to differentiate between genuine and generated instances. The GAN loss, vital for image reconstruction, is formally articulated as:

$$\mathcal{L}_{gan} = \mathbb{E}[\log(D_\phi(x))] + \mathbb{E}[\log(1 - D_\phi(G_\theta(\hat{z}_{on})))]. \tag{3}$$

In the given formulation, $D_\phi(x)$ represents the discriminator's estimated probability that the instance $x$ is sampled from the real data distribution. The generator, denoted by $G_\theta$, aims to produce instances that the discriminator $D_\phi$ perceives as real, maximizing the likelihood of them being classified as genuine. Conversely, the discriminator $D_\phi$ endeavors to distinguish real instances from those generated by $G_\theta$, classifying them accurately as either real or fake.

The objective function for disentangling ordinal content and non-ordinal information is combined with the GAN loss and the sparsity loss, i.e.,

$$\arg \min_{\{\phi,\theta,f_y,\boldsymbol{M}\}} \mathcal{L}_{\text{Aug}} = \arg \min_{\{\phi,\theta,f_y,\boldsymbol{M}\}} \mathcal{L}_{\text{gan}} + \lambda \cdot \mathcal{L}_{\text{sp}}. \tag{4}$$

where $\lambda$ is the coefficient to control the contribution of the sparsity loss to the overall objective.

### 3.3 CONTENT-PRESERVING AUGMENTATION FOR ORDINAL REGRESSION

Our method is crafted to complement existing ordinal regression techniques, leveraging the strengths of contrastive learning. After training, we could have a generator $G_{\hat{\theta}}$. To generate an instance $x'$ corresponding to a specific ordinal label, we employ Eq. 2. For instance, generating an example for the ordinal label $Y = 1$ can be achieved by:

$$\boldsymbol{x_i'} = G_{\hat{\theta}}(\hat{z}_{on}), \;\; \hat{z}_{on} = \hat{\boldsymbol{z}} + \boldsymbol{M} \odot f_{\boldsymbol{1}}(\hat{\boldsymbol{z}}), \;\; \hat{z} \sim \mathcal{N}(0, \mathbf{I}), \;\; Y = 1. \tag{5}$$

Specifically, we start by sampling $\hat{z}_{on}$. By setting the ordinal label $Y = 1$, we utilize $f_{\boldsymbol{1}}(\hat{\boldsymbol{z}})$ to generate $\hat{z}_{on}$ specific to the ordinal label $Y = 1$. By sampling different $\hat{z}_{on}$ values and maintaining the ordinal label $Y = 1$ constant, we can generate diverse instances that, while exhibiting stylistic variations, consistently belong to the label $Y$.

To integrate with existing ordinal regression methods, $x'$ can be employed as the strongly augmented data. We then incorporate the supervised contrastive loss (Khosla et al., 2020) as a regularization term for the prevailing method. This loss emphasizes intra-class similarities while concurrently maximizing inter-class disparities. Let's define $\mathbf{X}$ as the feature space of $x$. We introduce $h_\psi$ as a model with learnable parameters $\psi$, which is used by the ordinal regression method. For a given instance $x$, $h_\psi(x) = z$ outputs the latent representation $z$ used for ordinal regression. The contrastive loss on $h_\psi$ is defined as:

$$\mathcal{L}^{\text{con}} = \sum_{i \in I} \mathcal{L}_i^{\text{con}} = -\sum_{i \in I} \frac{1}{|\mathcal{S}(h_\psi(\boldsymbol{x_i}))|} \sum_{h_\psi(\boldsymbol{x_s}) \in \mathcal{S}(h_\psi(\boldsymbol{x_i}))} \log \frac{\exp\left(h_\psi(\boldsymbol{x_i}) \cdot h_\psi(\boldsymbol{x_i'})/\tau\right)}{\sum_{b \in B} \exp\left(h_\psi(\boldsymbol{x_i}) \cdot h_\psi(\boldsymbol{x_b})/\tau\right)}. \tag{6}$$

In the equation above, $I$ denotes the set of sample indices in the batch. For each instance $\boldsymbol{x_i}$ belonging to $y$, its strongly augmented counterpart $x_i'$ is generated using our method by setting the ordinal label $Y = y$ during the generation process.

## 4 EXPERIMENTS

In this section, we evaluate our method across three real-world applications within the domain of ordinal regression: age estimation, diabetic retinopathy rating, and weather condition prediction. Due to space constraints, we include qualitative analyses of our generative model in Appendix D.

**Baselines.** We employ five state-of-the-art deep learning-based ordinal regression methods as our baselines. OR-CNN (Niu et al., 2016) utilizes a series of binary classifiers and optimizes the model through the one-hot encoding of labels. CNNPOR (Liu et al., 2018b) reduces the multi-class negative log-likelihood while concurrently maintaining the intrinsic ordinal relationship among instances. SORD (Diaz & Marathe, 2019) employs a soft labeling strategy during training. POE (Li et al., 2021) captures data uncertainty via probabilistic embeddings. MWR (Shin et al., 2022) leverages an auxiliary set of reference images to model ordinal relationships. All these models are end-to-end trainable. We seamlessly integrate our contrastive learning objective into their original loss formulations, augmented with ordinal content-preserving data transformations.

**Experimental Settings.** For the generative model, we use StyleGAN2 (Karras et al., 2020) as the base model, $\lambda_1$ is set to 1e-4 across all settings. For all ordinal regression methods, we use VGG16 (Simonyan & Zisserman, 2014) the base deep neural network architecture, with ImageNet (Deng et al., 2009) pre-trained weight for initialization. We employ an embedding layer before the final output layer in the model to extract feature embeddings. The dimension of feature embedding is set to 128. The ratio of contrastive loss is consistently set to 1e-4 for OR-CNN, CN-NPOR and POE, and 1e-5 for SORE and MWR. For the three datasets, the input images are resized

| *Adience* | w/o OCP-CL | | w/ OCP-CL | |
|---|---|---|---|---|
| | Accuracy (↑) | MAE (↓) | Accuracy (↑) | MAE (↓) |
| OR-CNN (Niu et al., 2016) | $54.6 \pm 5.5$ | $0.60 \pm 0.09$ | $57.1 \pm 5.1$ *(+2.5)* | $0.56 \pm 0.06$ *(+0.04)* |
| CNNPOR (Liu et al., 2018b) | $55.1 \pm 6.0$ | $0.60 \pm 0.08$ | $57.7 \pm 4.2$ *(+2.6)* | $0.55 \pm 0.07$ *(+0.05)* |
| SORD (Diaz & Marathe, 2019) | $57.8 \pm 4.9$ | $0.53 \pm 0.06$ | $59.9 \pm 5.0$ *(+2.1)* | $0.49 \pm 0.06$ *(+0.04)* |
| POE (Li et al., 2021) | $60.5 \pm 4.8$ | $0.47 \pm 0.08$ | $63.7 \pm 4.6$ *(+3.2)* | $0.43 \pm 0.07$ *(+0.04)* |
| MWR (Shin et al., 2022) | $62.6 \pm 5.0$ | $0.45 \pm 0.08$ | $63.6 \pm 4.7$ *(+1.0)* | $0.43 \pm 0.07$ *(+0.02)* |

| *Diabetic Retinopathy* | w/o OCP-CL | | w/ OCP-CL | |
|---|---|---|---|---|
| | Accuracy (↑) | MAE (↓) | Accuracy (↑) | MAE (↓) |
| OR-CNN (Niu et al., 2016) | $71.9 \pm 1.3$ | $0.42 \pm 0.01$ | $72.8 \pm 0.7$ *(+0.9)* | $0.41 \pm 0.00$ *(+0.01)* |
| CNNPOR (Liu et al., 2018b) | $71.3 \pm 1.1$ | $0.42 \pm 0.02$ | $72.6 \pm 1.0$ *(+1.3)* | $0.41 \pm 0.01$ *(+0.01)* |
| SORD (Diaz & Marathe, 2019) | $69.1 \pm 1.0$ | $0.45 \pm 0.01$ | $69.9 \pm 1.1$ *(+0.8)* | $0.44 \pm 0.01$ *(+0.01)* |
| POE (Li et al., 2021) | $73.6 \pm 1.0$ | $0.40 \pm 0.01$ | $74.8 \pm 0.8$ *(+1.2)* | $0.38 \pm 0.00$ *(+0.02)* |
| MWR (Shin et al., 2022) | $74.5 \pm 1.1$ | $0.38 \pm 0.02$ | $75.1 \pm 1.1$ *(+0.6)* | $0.37 \pm 0.01$ *(+0.01)* |

| *SkyFinder* | w/o OCP-CL | | w/ OCP-CL | |
|---|---|---|---|---|
| | Accuracy (↑) | MAE (↓) | Accuracy (↑) | MAE (↓) |
| OR-CNN (Niu et al., 2016) | $60.3 \pm 2.1$ | $0.48 \pm 0.03$ | $62.1 \pm 2.3$ *(+1.8)* | $0.46 \pm 0.04$ *(+0.02)* |
| CNNPOR (Liu et al., 2018b) | $57.6 \pm 1.6$ | $0.52 \pm 0.03$ | $59.7 \pm 1.5$ *(+2.1)* | $0.49 \pm 0.03$ *(+0.03)* |
| SORD (Diaz & Marathe, 2019) | $58.2 \pm 1.9$ | $0.51 \pm 0.06$ | $60.5 \pm 2.0$ *(+2.3)* | $0.48 \pm 0.04$ *(+0.03)* |
| POE (Li et al., 2021) | $61.9 \pm 1.7$ | $0.46 \pm 0.05$ | $64.1 \pm 1.6$ *(+2.2)* | $0.42 \pm 0.05$ *(+0.04)* |
| MWR (Shin et al., 2022) | $62.4 \pm 1.8$ | $0.45 \pm 0.05$ | $63.2 \pm 1.9$ *(+0.8)* | $0.44 \pm 0.06$ *(+0.01)* |

Table 1: Accuracy (%) and MAE comparison on Adience dataset (Eidinger et al., 2014), Diabetic Retinopathy dataset (Liu et al., 2018a) and SkyFinder dataset (Mihail et al., 2016).

into $256 \times 256$ and center cropped into a sub-region of $224 \times 224$. Adam (Kingma & Ba, 2014) optimizer is used for all baseline methods, with a base learning rate of 1e-4. We uniformly train all baseline models for 200 epochs with a batch size of 256 for all baselines except MWR. For MWR, the batch size is set to 128 due to memory constraints. We report the results via the accuracy and mean absolute error (MAE) metrics. For the other parameters in the baselines, we adhere to the original settings designed in the papers unless specified in our experimental settings. While we employ our techniques for strong augmentations, weak augmentations are achieved solely through resizing, center cropping, and normalizing the original instances. No other augmentation methods are applied to the data. All experiments are conducted in on two 48GB NVIDIA RTX A6000 GPUs.

## 4.1 AGE ESTIMATION

**Dataset.** Age estimation is the task of predicting age groups based on facial images. The Adience dataset (Eidinger et al., 2014) comprises 26,580 photos from Flickr, featuring 2,284 subjects. These photos are annotated across eight age groups: 0-2, 4-6, 8-13, 15-20, 25-32, 38-43, 48-53, and over 60 years. The dataset adheres to a standard five-fold, subject-exclusive cross-validation protocol, as widely utilized in previous studies (Rothe et al., 2018; Shen et al., 2018; Li et al., 2019; 2021). The generative model is trained in accordance with the training fold of this protocol. For each instance in the dataset, we generate 3 augmented views using the generative model, the augmented views and the original instances are jointly trained by the models.

**Results.** We present the experimental results in Table 1 (Top). Employing our proposed OCP-CL method, OR-CNN experiences a 4.58% boost in accuracy and a 6.67% reduction in MAE. CNNPOR benefits from a 4.72% increase in accuracy and an 8.33% improvement in MAE. SORD's performance is uplifted by 3.63% in accuracy and 7.55% in MAE. POE sees the largest accuracy improvement of 5.29% and an MAE reduction of 8.51%. Lastly, MWR has a modest 1.6% increase in accuracy and a 4.44% decrease in MAE. This consistent improvement across multiple ordinal regression methods validate the efficacy of our OCP-CL approach for the task of age estimation. Additionally, we visualise the generative augmentations in Figure 1. We observe that the augmentations have preserved the ordinal content information for their respective age groups, capturing details such as the sparse eyebrows of children, silky skin texture of young adults, and the pronounced wrinkles of seniors, thereby allowing the ordinal regression methods and the contrastive learning framework to effectively learn the critical ordinal content information.

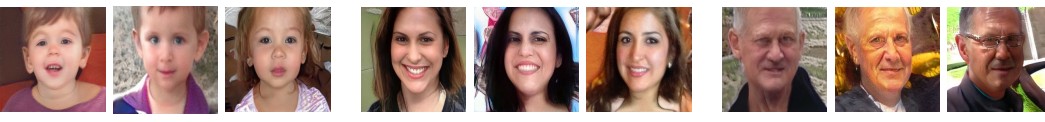

| Age Group: 4-6 | Age Group: 25-32 | Age Group: 60+ |

Figure 4: Generated augmentations for the age estimate task, the collections corresponds to the age group of (4-6), (25-32), and 60+ respectively.

## 4.2 DIABETIC RETINOPATHY RATING

**Dataset.** The Diabetic Retinopathy dataset[1] is utilized for predicting the severity stages of Diabetic Retinopathy based on high-resolution RGB retina images. The dataset consists of 35,126 individual instances, each annotated into one of five ordinal categories representing increasing levels of severity (*i.e.*, *No DR*, *Mild*, *Moderate*, *Severe*, and *Proliferative DR*). The dataset is partitioned into training, validation, and testing sets, which constitute 80%, 5%, and 15% of the total dataset, respectively. The dataset contains 25,810, 2443, 5292, 873 and 708 images for each category, respectively. Account for the imbalances between adjacent categories, we generate augmented views dynamically depending on the ground truth label, which mitigates the class imbalance issue. Specifically, the number of augmentations for instances from each increasing level of severity is set as [1, 3, 2, 5, 5].

**Results.** In Table 1 (Middle), we evaluate the performance of various ordinal regression methods on the Diabetic Retinopathy dataset, with and without the incorporation of our proposed OCP-CL (Ordinal Content-Preserving Contrastive Learning) module. Remarkably, all the compared methods exhibit improvement in both accuracy and MAE upon integration with the OCP-CL module. Practically, the incorporation of the OCP-CL module results in accuracy improvements of 1.25%, 1.82%, 1.16%, 1.63%, and 0.81% for OR-CNN, CNNPOR, SORD, POE, and MWR, respectively. Concurrently, the MAE reduces by 2.38%, 2.38%, 2.22%, 5.00%, and 2.63%, respectively. These results collectively indicate that the introduction of the OCP-CL module consistently enhances the performance across a diverse set of ordinal regression models. This validates the generalizability and efficacy of our proposed OCP-CL approach in boosting performance for ordinal regression tasks.

## 4.3 WEATHER CONDITION PREDICTION

**Dataset.** The SkyFinder Dataset (Mihail et al., 2016) comprises 94,804 labeled outdoor images, sourced from 53 static webcams affiliated with the Archive of Many Outdoor Scenes (AMOS). These images encapsulate a broad spectrum of weather and lighting conditions. A specialized subset of 62,988 images, specifically featuring the weather conditions of *Clear*, *Partly Cloudy*, and *Mostly Cloudy*, has been curated to create a weather prediction dataset. This subset is further partitioned into training, validation, and testing sets, constituting 80%, 5%, and 15% of the dataset, respectively. For each instance in the dataset, we generate 3 augmented views using the generative model, the augmented views and the original instances are jointly trained by the models.

**Results.** Table 1 (Bottom) presents the results of our experiments, highlighting the performance improvements achieved by all baseline models upon the incorporation of the contrastive module. Specifically, the accuracy improvements for the baselines are 2.9%, 3.5%, 3.8%, 3.4%, and 1.1% for OR-CNN, CNNPOR, SORD, POE, and MWR, respectively. Similarly, the improvements in MAE for the baselines are 4.2%, 5.8%, 5.9%, 8.7%, and 2.2%, respectively. With an average improvement of 2.94% in accuracy and 5.42% in MAE, these results demonstrate the efficacy of our method in enhancing the performance of deep-learning-based ordinal regression models on the weather condition estimation task.

## 4.4 ANALYSIS

**Transfer Learning.** In this section, we evaluate the transfer learning performance of our contrastive learning approach. Initially, we pre-train the encoder using a contrastive learning objective

---

[1] Accessible from https://www.kaggle.com/competitions/diabetic-retinopathy-detection

| Dataset | SupMoCo (He et al., 2019) | | SupCon (Khosla et al., 2020) | | S-LooC (Xiao et al., 2021) | | SupCReg (Zha et al., 2022) | | OCP-CL (Ours) | |
|---|---|---|---|---|---|---|---|---|---|---|
| | Accuracy | MAE | Accuracy | MAE | Accuracy | MAE | Accuracy | MAE | Accuracy ($\uparrow$) | MAE ($\downarrow$) |
| DR | 63.6 | 0.52 | 60.5 | 0.55 | 63.0 | 0.52 | 62.7 | 0.53 | **65.2** | **0.50** |
| Adience | 51.9 | 0.65 | 51.7 | 0.67 | 52.4 | 0.64 | 51.2 | 0.63 | **53.1** | **0.61** |
| SkyFinder | 56.1 | 0.55 | 53.9 | 0.59 | 55.6 | 0.56 | 55.1 | 0.55 | **57.5** | **0.52** |

Table 2: Linear evaluation on supervised contrastive learning frameworks. Accuracy (%) and MAE are reported for various ordinal datasets including Diabetic Retinopathy dataset, Adience (Levi & Hassner, 2015) and SkyFinder dataset (Mihail et al., 2016).

for feature extraction. Following this, we freeze the trained encoder and employ the extracted features as input to a single-layer MLP predictor, which is then fine-tuned on the training data. We assess the efficacy of our approach against recent state-of-the-art supervised contrastive learning frameworks across three different tasks. To mitigate performance degradation due to parameter settings, we dynamically adopt the recommended configurations from the original papers. However, for SupCon (Khosla et al., 2020), a batch size of 1024 is unfeasible for image instances of size 224 by 224. To ensure convergence, we reduced the image size to 64 by 64 and the batch size to 512. The results are presented in Table 2. Notably, our method significantly outperforms all existing approaches in the task of ordinal regression. The benefits of ordinal content-preserving data augmentation become evident when benchmarked against SupMoCo (He et al., 2019). Specifically, we adopt the SupMoCo framework as the baseline contrastive learning framework and integrate our augmentation strategy by replacing the original data augmentation modules. This aids in evaluating transfer learning performance. The contrastive loss formulation in SupMoCo aligns with our Eq. 6, and an additional momentum encoder is incorporated to ensure training convergence. The MoCo strategy is not employed in other experiments.

**Minimal Change in Image Generation.** We study the effect of minimal change on ordinal data generation by manipulating the mask hyperparameter. When this hyperparameter is set to zero, sparsity is not enforced, effectively removing the minimal change constraint from the generative process. As illustrated in Figure 5 ($\lambda_{mask} = 0 \rightarrow$ no minimal change), the absence of minimal change leads to the inclusion of non-ordinal features, such as hair, which do not contribute to identifying infant age groups and should be considered non-ordinal factors. However, in the case where minimal change is not applied, these features are learned as ordinal content factors and appear in all generated images, thereby demonstrating poor disentanglement performance. By enforcing minimal change, these non-ordinal elements are suppressed, enhancing the overall quality of the generated instances.

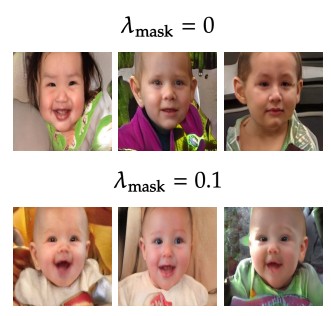

Figure 5: Influence of *minimal change* in image generation.

## 5 CONCLUSION

In this paper, we address the open challenge of applying contrastive learning to ordinal regression tasks. We find that the strong data augmentations in the contrastive learning frameworks often diminish the intrinsic discriminative semantic information associated with ordinal labels. Consequently, when contrastive learning is used to identify invariant features between weakly and strongly augmented views, the extracted features frequently lack the essential ordinal content information. To mitigate this issue, we introduce a novel augmentation method grounded in the principle of *minimal change*. This generative approach ensures that the images retain the essential ordinal content information during the data augmentation process. As a result, our method enhances the applicability of contrastive learning to ordinal regression tasks. Extensive experiments validate the efficacy of this approach in improving the performance of existing ordinal regression models. This work not only broadens the scope of contrastive learning in ordinal regression but also provides valuable insights for future research aimed at preserving crucial task-specific information during data augmentation.

## 6 ACKNOWLEDGEMENT

The authors would like to thank Muyang Li and Xiaobo Xia for their valuable feedback throughout the project. Jiyang Zheng is supported by the CSIRO Next Generation Graduates and AI for Missions PhD program. Tongliang Liu is partially supported by the following Australian Research Council projects: FT220100318, DP220102121, LP220100527, LP220200949, and IC190100031. BH was supported by the NSFC General Program No. 62376235, Guangdong Basic and Applied Basic Research Foundation Nos. 2022A1515011652, 2024A1515012399, HKBU Faculty Niche Research Areas No. RC-FNRA-IG/22-23/SCI/04, and HKBU CSD Departmental Incentive Scheme.

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

APPENDIX

## A    ABLATION ANALYSIS

**Number of Augmented Instances.**   We conduct an experiment to assess the sensitivity of the models to the number of augmented instances. Specifically, we adjust the number of views from the set [1,3,5,10] and "dynamic", and exam the performance on POE w/ OCP-CL. As illustrated in Figure 6, the models' performance remains relatively stable when the number of augmented instances ranges between 3 and 5. However, over-supplementing the data with augmented instances can lead to a degradation in model performance. Interestingly, we find that a dynamic number of augmentations depending on the class could benefit the models. This is particularly relevant for ordinal regression datasets that suffer from class imbalance. Specifically, by increasing the number of instances in underrepresented classes, we observe an improvement in the overall performance of the methods.

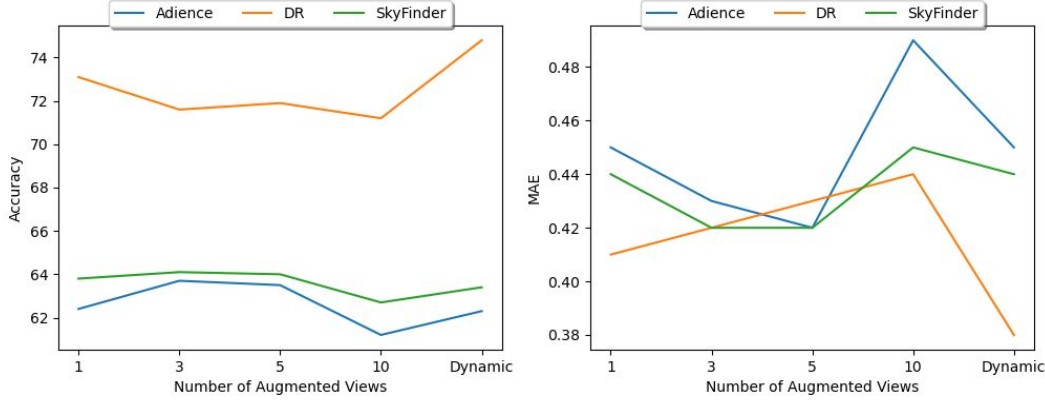

Figure 6: Ablation Study on the Number of Augmented Views.

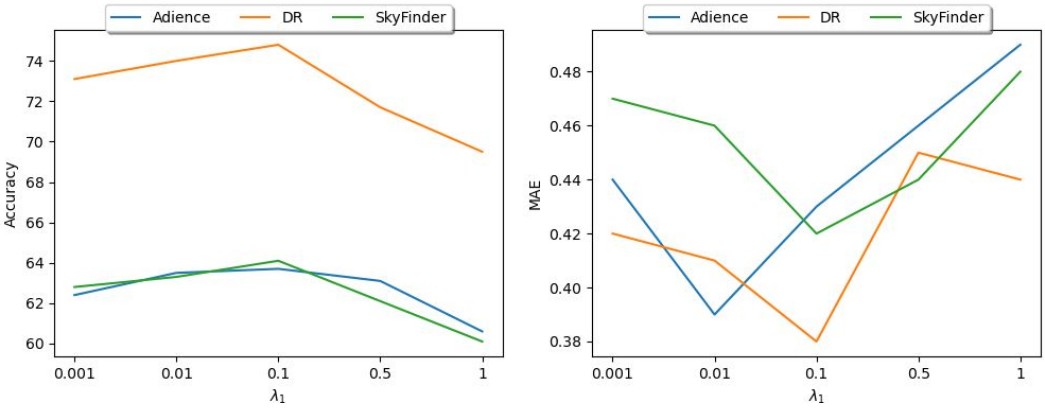

Figure 7: Sensitivity Analysis on $\lambda_1$ ratio.

**Sensitivity Analysis on $\lambda_1$.**   We conduct experiments to assess the impact of the mask sparsity ratio, denoted as $\lambda_1$, on the performance of the ordinal regression model across three downstream tasks. For this purpose, we utilize POE w/ OCP-CL to examine sensitivity to changes in $\lambda_1$. We

test five different sparsity ratios within the range of 1e-3 and 1. The results, presented in Figure 7, indicate that the ordinal regression model achieves optimal performance when $\lambda_1$ is set to 0.1 for all downstream tasks, and its performance decreases linearly with increases in the ratio beyond 0.1.

# B  ADDITIONAL RELATED WORKS

In this section, we provide additional background discussions relevant to our work. Specifically, we discuss recent advancements in Generative Data Augmentation, Disentangled Representation Learning, and Nonlinear ICA and explore their relationship with our research.

**Generative Models for Data Augmentations.** Instead of generating data augmentations using predefined transformations, Generative Data Augmentation (GDA) employs an alternative approach that leverages Deep Latent Variable Models (DLVMs) to generate new synthetic views from existing samples, based on conditional generative processes. Antoniou et al. (2017) and Tran et al. (2017) propose the use of Generative Adversarial Networks (GANs) (Goodfellow et al., 2014; Xia et al., 2022) to create a broader set of augmented data. More recently, Diffusion Models (DMs) (Sohl-Dickstein et al., 2015) have been utilized to alter high-level semantic attributes, thereby addressing the problem of lack of diversity along key semantic axes in data augmentation (Trabucco et al., 2023). While these approaches can generate impressive results that appear both realistic and novel, most of them are not guaranteed to maintain the invariance of the original data. Our proposed method fulfills the need for controllable generative data augmentation, offering a more trustworthy GDA approach.

**Disentangled Representation Learning.** The objective of Disentangled Representation Learning (DRL) is to construct a model proficient in recognizing and isolating the latent factors concealed within observable data (Wang et al., 2022). This isolation into semantically meaningful factors enhances the model's ability to produce interpretable data representations, thereby simulating the cognitive processes humans employ in understanding objects or relationships. In the context of generative modeling, Higgins et al. (2017) introduce a $\beta$-penalty coefficient for the KL divergence term in the evidence lower bound of a Variational Autoencoder (VAE) (Kingma & Welling, 2013; Li et al., 2022b; Huang et al., 2022; Hong et al., 2024; Lin et al., 2023) to balance latent channel capacity and independence constraints with reconstruction accuracy. Subsequently, various modifications to VAE have been introduced to improve its capability for disentanglement (Chen et al., 2018; Kumar et al., 2017). These include the incorporation of either implicit or explicit inductive biases as well as the utilization of diverse regularization techniques. On the other hand, InfoGAN (Chen et al., 2016) was the first to address the problem of disentangling latent factors in Generative Adversarial Networks, introducing an extra variational regularization of mutual information. Lin et al. (2019) introduced InfoGAN-CR, an unsupervised extension of InfoGAN that includes a contrastive regularizer to infer latent dimensions. Zhu et al. (2021) present PS-SC GAN, which builds upon InfoGAN and features a Spatial Constriction (SC) strategy to extract significant areas influenced by each latent dimension, along with a Perceptual Simplicity (PS) approach to make the latent factors more unambiguous. Wei et al. (2021) propose a method known as Orthogonal Jacobian Regularization (OroJaR) aimed at enhancing disentanglement in generative models. OroJaR uses the Jacobian matrix to examine how output alterations correspond to changes in input variables, specifically the latent dimensions. Our methods are parallel to GAN-based DRL methods, wherein we disentangle the latent factors by introducing the principle of minimal change.

**Nonlinear ICA.** Nonlinear independent component analysis (ICA) theoretically addresses the problem of disentangling latent factors when a nonlinear invertible transformation function exists, mapping independent samples to the latent space Hyvarinen & Morioka (2016; 2017). Recent developments Locatello et al. (2020); Zimmermann et al. (2021); Xie et al. (2022); Kong et al. (2022) indicate that, within a conditional generative process, the true latent factors might become identifiable when auxiliary information is provided. Khemakhem et al. (2020) demonstrate that the joint data and latent space distributions can be recovered, up to a simple transformation in the latent space, provided the generative process conditions on a variable observed alongside the data. Von Kügelgen et al. (2021) employ two views of the same image to disentangle the latent factors into ordinal content and non-ordinal components, with only the content component associating with the image's semantics. Our generative model leverages Nonlinear ICA theories to theoretically justify the disen-

tanglement of latent factors. By manipulating the ordinal content variable, while keeping the ordinal content factors consistent, our model can generate augmented views that preserve ordinal content.

## C   INTUITION OF WHY AUGMENTING NON-ORDINAL FACTORS

Here, we provide further discussion on why augmenting non-ordinal factors in images can benefit the training of ordinal regression/classfication models. In general, data augmentation aims to modify the styling factors in original examples that are not related to the predictive objectives of downstream tasks (Von Kügelgen et al., 2021).

In computer vision tasks, by changing the styles in images, we add more variety to the training data. This helps the model not to focus too much on the specific styles it sees in the training images. It teaches the model to recognize objects or features in images, no matter how the style of the image changes. This is important because in the real world, images can come in many different styles. So, adding style changes in training helps the model perform well on all kinds of images

In the context of ordinal regression, style information is referred to as non-ordinal information, governed by underlying non-ordinal factors. Our method also aims to enrich style diversity by altering non-ordinal information. This is achieved by randomly sampling non-ordinal factors while maintaining the ordinal content factors, then generating examples based on these factors. By preserving the ordinal content factors, we can change the image's style while keeping its ordinal content unchanged, thereby generating synthetic (counterfactual) images not seen in the training data. This approach enables neural networks to access more samples with diverse styles, thereby improving the generalization capabilities of ordinal regression models for unseen samples. As demonstrated in Figure 7, by randomly sampling non-ordinal factors while maintaining the ordinal content factors, we can alter various aspects of the image's style, such as people's dressing, background, and camera angles, etc., ensuring the ordinal content remains unchanged.

It is also important to emphasize two major advantages of our data augmentation methods: Firstly, existing image augmentation strategies do not guarantee the preservation of ordinal information during the augmentation process. For example, color jittering can change an image's color, potentially altering white hair to yellow, which could obscure the age of the person in the image. Secondly, our proposed data augmentation method is general. Our approach can be broadly applied to automatically infer ordinal content from other types of information and generate new examples with guarantees. While primarily tested on image data, our method's framework should be adaptable to non-image data. This adaptability is not achievable with traditional data augmentation methods, which mainly focus on image data. For instance, applying rotations to non-image data is not feasible.

## D   VISUALISATION OF DATA AUGMENTATION

We provide additional visualizations of the augmentation results. As shown in Figure 8, we conducted experiments on three ordinal regression datasets. Our findings indicate that our augmentation method effectively retains age-related and weather-related features, while simultaneously introducing significant stylistic variations. In the case of diabetic retinopathy instances, the differences between the original and augmented views are subtle. Without domain-specific knowledge, it is challenging to conclusively determine whether the ordinal content has been preserved. Figure 9 demonstrates the generative results for altering the ordinal factors. For each instance, we fix the non-ordinal factors and replace the ordinal factors with age-specific ordinal factors (i.e., representing different age groups). The age-specific ordinal factor is extracted from training images of the corresponding age group. By visualizing the results, we can observe that the age of the individuals has changed following augmentation, while the styling information from non-ordinal factors remains similar. This effectively illustrates the efficacy of our method in disentangling ordinal and non-ordinal factors. We also present image generation results from a conventional GAN (Karras et al., 2020) in Figure 10. It is important to note an advantage of our method over conventional GANs: the inability of conventional GANs to disentangle ordinal factors from non-ordinal factors. This means they cannot guarantee the preservation of an image's semantic information. Additionally, our model focuses more on fine-grained details when constructing novel samples. This is evident in the detailed modeling of age-related components in facial images. While conventional GANs

can achieve high generative quality, they sometimes fail to accurately represent certain age-related features in the images, such as generate hair for infants and child face for seniors.

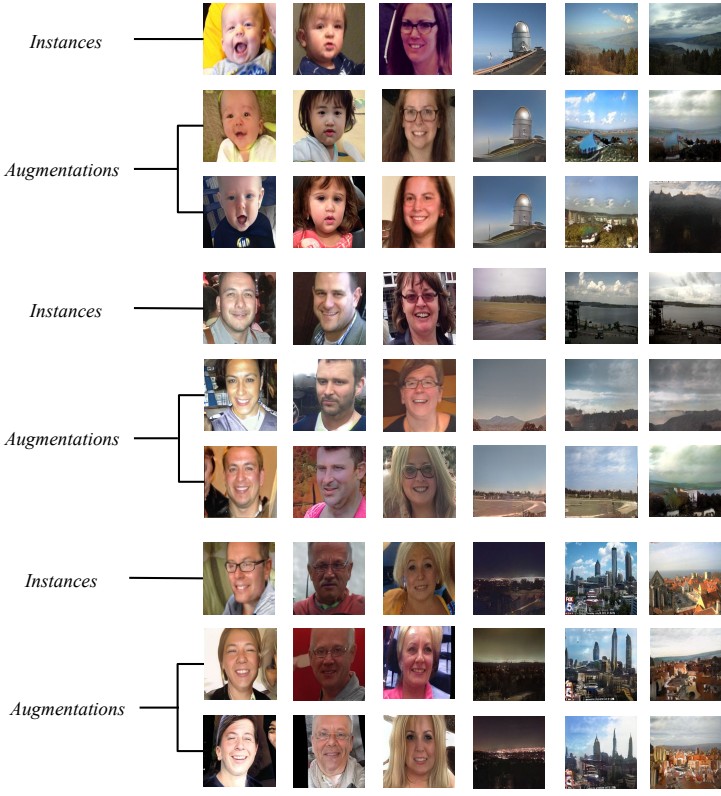

Figure 8: Generated augmentations by augmenting the non-ordinal factors $\hat{z}_n$.

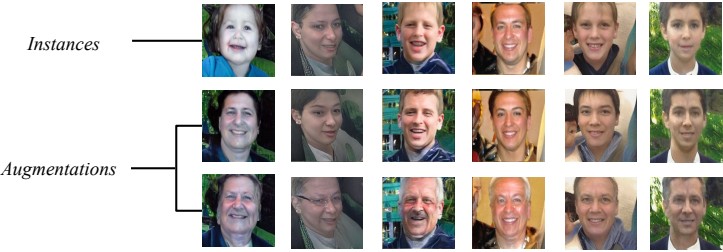

Figure 9: Generated augmentations by augmenting the ordinal factors $\hat{z}_o$ with age-specific factors.

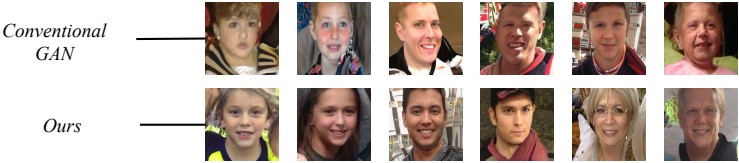

Figure 10: Unconditional image generation results of conventional GAN (the first Row) and Our method (the second Row).

