# OpenReview forum: "Enhancing Contrastive Learning for Ordinal Regression via  Ordinal Content Preserved Data Augmentation"
_ICLR.cc/2024/Conference — ICLR 2024 poster_

### Official Review · Reviewer_yfaW · 2023-10-24

**Soundness:** 2 fair
**Presentation:** 3 good
**Contribution:** 3 good
**Rating:** 5
**Confidence:** 4

**Summary:**

The article proposes to apply contrastive learning to the ordinal regression problem and suggests that strong augmentation could eliminate task-related features when generating augmented images. To address this problem, the authors propose a plug-and-play method based on the principle of "minimal change" to generate the augmented images via GAN. This method could retain the desired ordinal information and consistently improve the performance of existing state-of-the-art methods in ordinal regression tasks.

**Strengths:**

1.The article points out the possible reliance on detail features in ordinal regression.
2.The authors propose to generate augmented images via GAN with the guidance of “minimal change”. The augmented images are applied in contrastive learning to boost performance for ordinal regression.
3.Experiments on multiple datasets demonstrate the improvements in performance.

**Weaknesses:**

1.The quality of the writing needs to be improved, and some important concepts are not explained.
(1)“Deep Sigmoid Flow” is not explained in detail. By the way, in the reference paper, there is only “deep sigmoidal flows”. Is "Deep Sigmoid Flow" a typo, or is it a completely different concept than “deep sigmoidal flows”?
(2)The detailed implementation of mask is not explained. I think it's important to clarify whether it's a category-wise operation or not.
2.The experiments are not sufficient.
(1)Image augmentation is not necessary in supervised contrastive learning. The article lacks comparisons with contrastive learning without image augmentation and with contrastive learning using conventional image augmentation. The benefits of additional image augmentation are unclear.
(2)The article does not conduct experiments based on conventional GANs and does not demonstrate the superiority of the proposed generation methods.
3.The proposed method is only appropriate for data augmentation in the supervised scenario, while most of the contrastive learning methods that have a strong dependence on augmentation are unsupervised.

**Questions:**

1.What is the difference between the proposed generative method and [1]?
2.How does minimal change ensure that content information related to ordinal is preserved?
3.Is the mask operation category-wise? if not, how does the model select whether or not to treat hair as ordinal information based on different age groups?
4.Can you provide more generated images? The two sets of images in Figure 5 are not from the same age group. In real life, most infants do not have hair and most children have. I think that if hair is not ordinal information, it is more beneficial to improve the performance of the classifier if part of the generated images have hair while others do not have hair. What's your opinion?
[1]Shaoan Xie, Lingjing Kong, Mingming Gong, and Kun Zhang. Multi-domain image generation and translation with identifiability guarantees. In The Eleventh International Conference on Learning Representations, 2022.

---

> ### Author Response · Authors · 2023-11-19
> **Responses from the authors - Part I**
>
> **We thank the reviewer for providing these valuable comments.**
>
> Below, we provide detailed responses to address each of the concerns raised.
>
> ---
>
> **Q1. “Deep Sigmoid Flow” is not explained in detail. By the way, in the reference paper, there is only “deep sigmoidal flows”. Is "Deep Sigmoid Flow" a typo, or is it a completely different concept than “deep sigmoidal flows”?**
>
> We are sorry for the confusion. The terms 'deep sigmoid flow' and 'deep sigmoidal flow' refer to the same concept. There has been a mix of usage in the related works (Kong et al. 2022, Xie et al., 2022).
>
> Deep sigmoidal flow (DSF) is a type of normalization flow characterized by the use of small neural networks with sigmoid units. These units introduce inflection points in the transformation function, enabling the modeling of complex probability distributions. Within our context, it serves as a component-wise transformation function. However, it's worth noting that other functions meeting the same criteria could also be employed in its place, but we have just followed the prior implementation of using DSF.
>
> ---
>
> **Q2. Image augmentation is not necessary in supervised contrastive learning.**
>
> Thank you for your insights, We agree with the observation that the performance enhancement from strong image augmentation in supervised contrastive learning may not be as important as in unsupervised learning.
>
> However, strong augmentations remain advantageous in supervised contrastive learning contexts. As evidenced in previous studies (Khosla et al., 2020), omitting strong image augmentations leads to noticeable performance degradation. To further demonstrate the value of data augmentation in our specific task, we will include a comparative analysis in the updated version of our paper. This analysis will contrast the performance of contrastive learning with and without data augmentations.
>
> ---
>
> **Q3. The article does not conduct experiments based on conventional GANs and does not demonstrate the superiority of the proposed generation methods.**
>
> Thank you for the suggestions. We will include comparison to conventional GANs in experiments for the updated version.
>
> ---
>
> **Q4. The proposed method is only appropriate for data augmentation in the supervised scenario, while most of the contrastive learning methods that have a strong dependence on augmentation are unsupervised.**
>
> Our focus is on the problem of ordinal regression, typically involving pairs of images and labels. A significant challenge in this domain is the subtlety of features in the original images that are crucial for determining ordinal classes. These features can become distorted through strong data augmentations in contrastive learning. Hence, our proposed method is specifically designed to tackle this issue in the context of ordinal regression, rather than addressing challenges in arbitrary unsupervised learning scenarios.

---

> ### Author Response · Authors · 2023-11-19
> **Responses from the authors - Part II**
>
> **Q5. What is the difference between the proposed generative method and (Xie et al., 2022)?**
>
> We are sorry for the confusion.
>
> The existing theoretical framework on latent factor disentanglement primarily falls into two categories: the first involves nonlinear ICA with auxiliary variables (Hyvärinen et al., 2019; von Kügelgen et al., 2021), and the second leverages the principle of minimal change (Kong et al., 2022; Xie et al., 2022). These methods (Kong et al., 2022; Xie et al., 2022) typically assume a specific data generative process and utilize generative models to model the generative process.
>
> However, in various scenarios, determinations regarding which factors should change and which should remain static vary. For instance, under the minimal change framework, Xie et al. (2022) developed a multi-domain image generation and translation model. In this model, they concluded that the influence of domain information should vary across different domains but have minimal impact on the image's distribution. Information that is critical for predicting labels, however, should stay constant across domains. Consequently, their proposed generative method adopts the principle of minimal change to limit the influence of domain shifts on style changes.
>
> In the context of ordinal regression, we propose that ordinal factors should change across different ordinal labels, yet exert minimal influence on the image's distribution. Other factors should remain constant. Inspired by this, we adopt the minimal change concept and demonstrate its effectiveness in disentangling ordinal content from non-ordinal factors under reasonable assumptions.
>
> Additionally, it is important to note the generality of our proposed data augmentation method. Our approach can be broadly applied to distinguish ordinal content from other information types and to generate new examples accurately. While it was primarily tested on image data, our method is conceptually adaptable to non-image data as well, offering guarantees that are not typical of other data augmentation methods. Intuitively, existing augmentations usually change factors like position, rotation, and color. However, these factors are limited to image data, and it is unknown how to perform augmentation on other types of data. Our method takes it a step further; it can automatically infer latent ordinal content and other non-ordinal factors. By intervening on the inferred non-ordinal factors, new examples can be consistently generated that retain the essential ordinal content information. Thus, our method is general and helps improve the reliability of current machine learning methods.
>
> ---
>
> **Q6.  How does minimal change ensure that content information related to ordinal is preserved?**
>
> Our method is grounded in the theoretical framework of minimal change (Kong et al. 2022), which is instrumental in achieving disentanglement. According to the theoretical insights from prior research, our method's applicability to ordinal regression becomes feasible. The principle implies that, given a sufficient number of images from different classes, the component $\hat{z}_o$ can be identified on a component-wise basis. This indicates that $\hat{z}_o$ undergoes a simple component-wise transformation solely from $z_o$. Consequently, $\hat{z}_o$ is devoid of information about the style component $z_n$, as it remains unaffected by $z_n$, thus achieving disentanglement.
>
> The following provides the intuitive explanation about disentanglement:
>
> - We categorize the latent factors into two groups: ordinal ($\hat{z}_o$) and non-ordinal ($\hat{z}_n$). In this framework, $\hat{z}_o$ serves as the repository for crucial ordinal content information that determines the ordinal category.
>
> - For reconstruction, we utilize both the predicted ordinal factors ($\hat{z}_o$) and the predicted non-ordinal factors ($\hat{z}_n$). $\hat{z}_o$ is specifically employed for predicting ordinal labels and is subjected to minimal change, restricting its influence in generating new instances.
>
> - Assuming the generative model attains minimal reconstruction error, $\hat{z}_o$, the factor designated for ordinal label prediction, effectively captures the ordinal content. By constraining it through the principle of minimal change, it encapsulates only the vital information pertinent to ordinal labels, while $\hat{z}_n$ accommodates other types of information.
>
> - This approach leads to a clear disentanglement of ordinal and non-ordinal information, enhancing the model's effectiveness.

---

> > ### Author Response · Authors · 2023-11-22
> > **Rolling discussion coming to an end – awaiting your valuable feedback**
> >
> > Dear Reviewer yfaW,
> >
> > Thank you for your insightful questions. As the rolling discussion period for our paper is coming to a close, we are still waiting for your feedback. Please do not hesitate to contact us, if you need further clarification on any aspect of the paper. We are more than happy to address any questions or concerns you may have to ensure a smooth review process.
> >
> > Warm regards,
> >
> > Authors

---

> ### Author Response · Authors · 2023-11-19
> **Responses from the authors - Part III**
>
> **Q7.Is the mask operation category-wise? If not, how does the model select whether or not to treat hair as ordinal information based on different age groups?**
>
> As shown in Equation 2, $f_y$ is a category-specific transformation function, encoding categorical information. A uniform mask is applied, aiming to learn which latent dimensions correspond to ordinal content factors and which to non-ordinal factors under the minimal change constraint.
>
> In the following, we provide a detailed explanation of the operations performed in Equation 2 for further clarification.
>
> In Equation 2, our approach starts by projecting the input latent variable $\hat{z}$ into a category-specific distribution, achieved through the transformation function $f$. This results in a projected latent variable that is distinctly different from the input and is specific to a given category.
>
> Next, we aim to isolate the dimensions associated with ordinal content. To accomplish this, we employ a mask operator, denoted as $M$. This operator is a learnable matrix, designed to match the shape of the input noise. Our adherence to the principle of minimal change is reflected in the way we constrain the mask $M$ to be sparse, ensuring that non-ordinal dimensions in the product are set to zero. We calculate the Hadamard product of $M$ with the projected input, effectively isolating the relevant dimensions.
>
> The final step involves reintegrating style information. This is done to compensate for the zeroed elements in the latent variables. Consequently, the resulting processed latent variables, designated as $\hat{z}_{on}$, are effectively disentangled, separating ordinal content from other information.
>
> ---
>
> **Q8.Can you provide more generated images? The two sets of images in Figure 5 are not from the same age group. In real life, most infants do not have hair and most children have. I think that if hair is not ordinary information, it is more beneficial to improve the performance of the classifier if part of the generated images have hair while others do not have hair. What's your opinion?**
>
>
> We have provided additional generated images in the supplementary materials.
>
> It is important to clarify that Figure 5 demonstrates the effect of disentanglement using the principle of minimal change. The two sets of images shown are from the same age group. With the influence of minimal change, the model preserves the 'no hair' feature for children, as evident in the second row. However, in the absence of this constraint, it becomes challenging to identify which dimensions should remain unchanged. Consequently, without applying the minimal change principle, 'hair' is incorrectly added to children in the post-augmentation images.
>
> ---
>
> **References**
>
> Hyvarinen, A., Sasaki, H., & Turner, R. (2019). Nonlinear ICA Using Auxiliary Variables and Generalized Contrastive Learning. In K. Chaudhuri & M. Sugiyama (Eds.), Proceedings of the Twenty-Second International Conference on Artificial Intelligence and Statistics (Vol. 89, pp. 859–868). PMLR.
>
> Khosla, P., Teterwak, P., Wang, C., Sarna, A., Tian, Y., Isola, P., Maschinot, A., Liu, C., & Krishnan, D. (2020). Supervised contrastive learning. Advances in Neural Information Processing Systems, 33, 18661–18673.
>
> Kong, L., Xie, S., Yao, W., Zheng, Y., Chen, G., Stojanov, P., Akinwande, V., & Zhang, K. (2022). Partial disentanglement for domain adaptation. In K. Chaudhuri, S. Jegelka, L. Song, C. Szepesvari, G. Niu, & S. Sabato (Eds.), Proceedings of the 39th International Conference on Machine Learning (Vol. 162, pp. 11455–11472). PMLR.
>
> Von Kügelgen, J., Sharma, Y., Gresele, L., Brendel, W., Schölkopf, B., Besserve, M., & Locatello, F. (2021). Self-supervised learning with data augmentations provably isolates content from style. Advances in Neural Information Processing Systems, 34, 16451–16467.
>
> Xie, S., Kong, L., Gong, M., & Zhang, K. (2022). Multi-domain image generation and translation with identifiability guarantees. The Eleventh International Conference on Learning Representations.

---

> ### Author Response · Authors · 2023-11-22
> **Summary of Our Response and Changes in Paper According to Your Comments**
>
> Dear Reviewer yfaW,
>
> We recognize that our response may be extensive in terms of reading length. To facilitate a quicker understanding, we have prepared a summary of our response, aligning with your comments and our changes. We hope this summary aids in efficiently conveying how we have addressed your concerns.
>
> For the main contents:
>
> - Comparison with Xie et al. (2022): We have discussed our approach's distinctions compared to their work.
>
> - Preservation of Content Information: We have further elaborated how minimal changes ensure the preservation of content information related to ordinal factors.
>
> - Details on Mask Operation: More comprehensive details about the mask operation have been provided.
>
> - Deep Sigmoidal Flow Explanation: Additional explanations have been provided for a better understanding of the Deep Sigmoidal flow.
> - Supervised contrastive learning with/without strong augmentation.

---

> ### Author Response · Authors · 2023-11-22
> **Additional Experiments for Q2 and Q3**
>
> For **Q2**:
> - Data augmentation for Supervised Contrastive Learning: **We conducted experiments on supervised contrastive learning without data augmentation for the POE ordinal regression model across three downstream tasks.** The results (Accuracy/MAE), as shown in the table below, indicate a negative impact on performance without appropriate data augmentation.
>
> |           | POE + CL w/o Augmentations | POE w/ OCP-CL |
> |-----------|:----------------------------:|:---------------:|
> | Adience   | 61.4/0.47                  | 63.7/0.43     |
> | DR        | 72.4/0.43                  | 74.8/0.38     |
> | SkyFinder | 61.7/0.46                  | 64.1/0.42     |
> ||
>
> ---
> For **Q3**
> - **We have included additional generated images in our appendix to enhance our findings**. These include:
>
> 1. Comparative Generative Results: Visual comparisons between the generative results of our method and conventional GAN models.
>
> > It is important to note an advantage of our method over conventional GANs: the inability of conventional GANs to disentangle ordinal factors from non-ordinal factors. This means they cannot guarantee the preservation of an image’s semantic information. Additionally, our model focuses more on fine-grained details when constructing novel samples. This is evident in the detailed modeling of age-related components in facial images. While conventional GANs can achieve high generative quality, they sometimes fail to accurately represent certain age-related features in the images, such as generating hair for infants and a child face for seniors.
>
> 2. Validation of Disentanglement: Visualizations demonstrating how changes in ordinal factors alter the ordinal categories of the images, thereby validating the disentanglement of ordinal content from non-ordinal factors. These visualizations aim to provide clearer insights and validation of the methodologies discussed in our paper.
>
> > For each instance, we fix the non-ordinal factors and replace the ordinal factors with age-specific ordinal factors (i.e., representing different age groups). The age-specific ordinal factor is extracted from training images of the corresponding age group. By visualizing the results, we can observe that the age of the individuals has changed following augmentation, while the styling information from non-ordinal factors remains similar. This effectively illustrates the efficacy of our method in disentangling ordinal and non-ordinal factors.
>
> The detailed explanations are located in the above sections and corresponding parts of the paper. We hope these points we have mentioned cover all of your concerns. Please let us know if you still have any other questions.

---

> ### Author Response · Authors · 2023-11-22
> **Awating for Your Valuable Feedback**
>
> Dear Reviewer yfaW,
>
> We appreciate the valuable comments and insights you have provided on our paper. In response to your feedback, we have made efforts to address all of your concerns comprehensively.
>
> The rolling discussion period will be closed soon, we would like to hear your feedback.
>
> Could you please let us know if there are any questions or aspects of our paper requiring further clarification? We are more than willing to provide the necessary information and details.
>
> Warm regards,
>
> The Authors

---

> ### Comment · Reviewer_yfaW · 2023-11-23
> **Response**
>
> Thanks for the response. I have fully read the rebuttal. Most of my concerns are solved, but there are still a few concerns are not fully addressed: the comparisons between the proposed method and GAN is insufficient, e.g., only comparisons based on the generated images are insufficient, and if the authors can provide comparisons of methods towards the ultimate model performance, it would be helpful; another concerns is that how the class-agnostic mask handles the problem of ordinal content information in some age groups but not ordinal content information in some age groups is not explained clearly. Concretely, the responses of authors addressed most of my concerns, but I still think the proposed method is not concise and ··elegant‘’ enough, and the intuition behind behaviours of the proposed method is not very solid. Thus, I would like to raise my score to weak reject.

---

> > ### Author Response · Authors · 2023-11-23
> > **Responses from the authors and Thank you for the feedback**
> >
> > Dear Reviwer yfaw,
> >
> > Thank you for your feedback on our rebuttal. Below, we provide the responses to address your last concerns.
> >
> > ---
> >
> > 1. More comparisons with conventional GAN
> >
> > Thank you for the suggestion. We will provide quantitative analysis in the future version of our paper, such as using FID scores, to further compare the generative quality with other conventional generative methods.
> >
> > One aspect we wish to emphasize is that the primary purpose of our generative method is not only to generate high-quality synthetic images. Although our proposed method demonstrates improved generative quality of ordinal data compared to conventional GANs, our main motivation is still to generate ordinal content preserving augmentations for enhancing ordinal regression model methods, a task that conventional GANs or other conventional generative models cannot accomplish.
> >
> > ---
> >
> > 2. How the class-agnostic mask handles the problem of ordinal content information in some age groups but not ordinal content information in some age groups is not explained clearly.
> >
> > The ordinal content information is handled by the category-specific transformation function $f_y(\cdot)$, where the specific $y$ is provided when doing augmentation. The mask operation will preserve such categorical information that differs from groups to groups.
> >
> > ---
> >
> > Due to the limited time before the close of the rolling discussion, we might not be able to elaborate in detail on our explanations. However, we will update future versions of our paper in accordance with your comments if possible.
> >
> > Again, we appreciate your valuable comments!
> >
> > Wish you a nice day,
> >
> > The Authors

---

### Official Review · Reviewer_gbdG · 2023-10-28

**Soundness:** 2 fair
**Presentation:** 3 good
**Contribution:** 3 good
**Rating:** 6
**Confidence:** 4

**Summary:**

In this paper, the authors concentrate on the ordinal regression task. They address the challenge that strong augmentations can often overshadow or dilute the ordinal content information. To mitigate this issue, they propose an augmentation method based on the principle of minimal change to replace the predefined strong augmentations.

**Strengths:**

1.The author has provided a clear statement of the paper's motivation.

2.The paper is well-organized.

**Weaknesses:**

1.The experimental results appear to be based on a single run, and the author should consider conducting multiple experiments to reduce the influence of randomness. Notably, the performance on the MWR in Table 1 seems to exhibit minimal variation. The author should provide clarification regarding whether this consistency is a result of chance or if there are specific underlying reasons. Additionally, the author should explore novel ways to demonstrate the effectiveness of the proposed method.

2.Further clarification is needed for the meaning of "M" in Formula 2. The specific process of obtaining "zo" through Formula 2 also requires detailed explanation, as this is a critical aspect that is currently lacking in the current version.

3.Formula 4 in the paper lacks sensitivity analysis for "lambda1."

4.Ablation experiments should be conducted to effectively demonstrate the impact and effectiveness of the proposed method.

**Questions:**

See weakness.

---

> ### Author Response · Authors · 2023-11-19
> **Responses from the authors - Part I**
>
> **We thank the reviewer for providing these valuable comments.**
>
> Below, we provide detailed responses to address each of the concerns raised.
>
> ---
>
> **Q1. The experimental results appear to be based on a single run, and the author should consider conducting multiple experiments to reduce the influence of randomness. Notably, the performance on the MWR in Table 1 seems to exhibit minimal variation. The author should provide clarification regarding whether this consistency is a result of chance or if there are specific underlying reasons. Additionally, the author should explore novel ways to demonstrate the effectiveness of the proposed method.**
>
> We would like to clarify that the results presented are derived from multiple runs or cross-validation, and we will include the standard deviations in the updated version. Regarding performance, we observed that MWR (Shin et al., 2022) shows relatively less improvement compared to other ordinal regression methods.
>
> Our hypothesis for this observation is that the ranking module in MWR inherently incorporates elements of contrastive training. This inherent feature possibly diminishes the benefits of explicit contrastive representation learning. Specifically, MWR's objective aligns with a generalized form of supervised contrastive regression (Zha et al., 2023), utilizing label orders to construct positive and negative pairs for training.
>
> In addition, we have rigorously evaluated the model's performance across a variety of benchmark datasets, consistently observing improvements in performance across all datasets and for every model variant tested. This also highlights our significant contribution to the field of ordinal regression.
>
> ---
>
> **Q2. Further clarification is needed for the meaning of "M" in Formula 2. The specific process of obtaining "zo" through Formula 2 also requires a detailed explanation, as this is a critical aspect that is currently lacking in the current version.**
>
> In Formula 2, our approach starts by projecting the input latent variable $\hat{z}$ into a category-specific distribution, achieved through a label specific transformation function $f_y$. This results in a projected latent variable $f_y(\hat{z})$ that is distinctly different from the input $\hat{z}$ and is specific to a given category.
>
> Next, we aim to isolate the dimensions associated with ordinal content. To accomplish this, we employ a mask operator, denoted as $M$. This operator is a learnable matrix, designed to match the shape of the input noise. Our adherence to the principle of minimal change is reflected in the way we constrain the mask $M$ to be sparse, ensuring that non-ordinal dimensions in the product are set to zero. We calculate the Hadamard product of $M$ with the projected input (e.g., $M \odot f_y(\hat{z})$), effectively isolating the relevant dimensions.
>
> The final step involves reintegrating style information. This is done to compensate for the zeroed elements in the latent variables. Consequently, the resulting processed latent variables, designated as $\hat{z}_{on}$, are effectively disentangled, separating ordinal content from other information.
>
> Our method is grounded in the theoretical framework of minimal change (Kong et al. 2022), which is instrumental in achieving disentanglement with theoretical guarantees. Specifically, the principle implies that, given a sufficient number of images from different classes, the component $\hat{z}_o$ can be identified on a component-wise basis. This indicates that $\hat{z}_o$ undergoes a simple component-wise transformation solely from $z_o$. Consequently, $\hat{z}_o$ is devoid of information about the style component $z_n$, as it remains unaffected by $z_n$, thus achieving disentanglement.
>
> ---
>
> References
>
> Shin, N.-H., Lee, S.-H., & Kim, C.-S. (2022). Moving window regression: A novel approach to ordinal regression. Proceedings of the IEEE/CVF Conference on Computer Vision and Pattern Recognition, 18760–18769.
>
>
> Zha, K., Cao, P., Son, J., Yang, Y., & Katabi, D. (2023). Rank-N-Contrast: Learning Continuous Representations for Regression. Thirty-Seventh Conference on Neural Information Processing Systems.
>
> Kong, L., Xie, S., Yao, W., Zheng, Y., Chen, G., Stojanov, P., Akinwande, V., & Zhang, K. (2022). Partial disentanglement for domain adaptation. In K. Chaudhuri, S. Jegelka, L. Song, C. Szepesvari, G. Niu, & S. Sabato (Eds.), Proceedings of the 39th International Conference on Machine Learning (Vol. 162, pp. 11455–11472). PMLR.

---

> ### Author Response · Authors · 2023-11-19
> **Responses from the authors - Part II**
>
> **Q3. Formula 4 in the paper lacks sensitivity analysis for "lambda1."**
>
> Thank you for highlighting this aspect. We have updated the paper with sensitivity analysis for $\lambda_1$ in the appendix section.
>
> ---
>
> **Q4. Ablation experiments should be conducted to effectively demonstrate the impact and effectiveness of the proposed method.**
>
> Thank you for your valuable suggestion.
>
> - We have incorporated a sensitivity analysis of $\lambda_1$ into our ablation study.
>
> - To demonstrate the efficacy of our proposed generative method, we have included a comparative analysis with conventional GANs.

---

> ### Author Response · Authors · 2023-11-22
> **Awating for Your Valuable Feedback**
>
> Dear Reviewer gbdG,
>
> We appreciate the valuable comments and insights you have provided on our paper. In response to your feedback, we have made efforts to address all of your concerns comprehensively.
>
> The rolling discussion period will be close soon, we would like to hear your feedback.
>
> Could you please let us know if there are any questions or aspects of our paper requiring further clarification. We are more than willing to provide the necessary information and detail.
>
> Warm regards,
>
> The Authors

---

> ### Author Response · Authors · 2023-11-22
> **Additional Experimental results Required by Q4**
>
> - **We have conducted a sensitivity analysis for $\lambda_1$ and included the analysis in the appendix.** We test five different sparsity ratios within the range of 1e-3 to 1 for POE w/ OCP-CL for downstream tasks. The results indicate that the ordinal regression model achieves optimal performance when $\lambda_1$ is set to 0.1 for all downstream tasks, and its performance decreases linearly with increases in the ratio beyond 0.1.
>
> - **We have conducted visual ablation studies on the disentanglement performance and generative quality of our proposed method**:
>
>   - For each instance, we fix the non-ordinal factors and replace the ordinal factors with age-specific ordinal factors (i.e., representing different age groups). The age-specific ordinal factor is extracted from training images of the corresponding age group. By visualizing the results, we can observe that the age of the individuals has changed following augmentation, while the styling information from non-ordinal factors remains similar. This effectively illustrates the efficacy of our method in disentangling ordinal and non-ordinal factors.
>
>   - We also present image generation results from a conventional GAN to compare with our proposed method. It is important to note an advantage of our method over conventional GANs: the inability of conventional GANs to disentangle ordinal factors from non-ordinal factors. This means they cannot guarantee the preservation of an image's semantic information. Additionally, our model focuses more on fine-grained details when constructing novel samples. This is evident in the detailed modeling of age-related components in facial images. While conventional GANs can achieve high generative quality, they sometimes fail to accurately represent certain age-related features in the images, such as generating hair for infants and a child's face for seniors.
>
> ---
>
> The detailed explanations are located in the above sections and corresponding parts of the paper. We hope these points we have mentioned cover all of your concerns. Please let us know if you still have any other questions.

---

> ### Author Response · Authors · 2023-11-22
> **Dear Reviewer gbdG, please let us know if you have any further concerns, thanks**
>
> Dear Reviewer gbdG,
>
> As the rolling discussion period for our paper is coming to a close, we are still waiting for your feedback. If you need further clarification on any aspect of the paper, please do not hesitate to contact us. We are more than happy to address any questions or concerns you may have to ensure a smooth review process.
>
> Warm regards,
>
> Authors

---

> > ### Author Response · Authors · 2023-11-23
> > **Thank you for your Valuable Comments**
> >
> > Dear Reviewer gbdG,
> >
> > We hope that all your concerns have been properly addressed. We wish to express our sincere gratitude for your effort of reviewing our paper and for providing such valuable feedback.
> >
> > Warm regards,
> >
> > The Authors

---

### Official Review · Reviewer_PxYT · 2023-11-06

**Soundness:** 3 good
**Presentation:** 2 fair
**Contribution:** 2 fair
**Rating:** 6
**Confidence:** 3

**Summary:**

In this paper, the authors aim to improve contrastive learning’s utility for ordinal regression. They find that strong data augmentation could lead to distortion of certain discriminative information (content information) in the image, which is crucial for ordinal regression. To this end, they propose to use a generative model to create augmented images, which diverse in different styles but have the same content information as the original image. Experimental results show that this approach enhance the performance of existing methods.

**Strengths:**

The motivation of this work is clear and reasonable, which provides a valid approach to tackle the problem of potential negative effects on performance caused by excessive augmentation. The experiments are sufficient to support the effectiveness of this method.

**Weaknesses:**

- The improvement in effectiveness comes at the cost of increasing computational overhead. The time and space required to train the generated model is no less than (or even greater than) that of training the ordinal regression model, yet the performance gain is not so significant.
- It is unclear whether the proposed framework could guarantee that to what extent the content information can be maintained in the invariant ordinal content factors $\hat{\tilde{z}}_O$, which is crucial for the quality of generated images and further affect the performance of the ordinal regression model.

**Questions:**

Q1. Do the performance of the model sensitive to the setting of $\lambda_1$ in Eq. (4)?

Q2. The authors choose GAN as the generative model. It seems that the proposed data generative process could also be implemented by other model families (e.g., VAE or diffusion models).

Q3. Could you explain that to what extent the content information can be maintained in $\hat{\tilde{z}}_O$, either in theory or experiments?

---

> ### Author Response · Authors · 2023-11-19
> **Responses from the authors - Part I**
>
> **We thank the reviewer for providing these valuable comments.**
>
> Below, we provide detailed responses to address each of the concerns raised.
>
> ---
>
> **Q1. The improvement in effectiveness comes at the cost of increasing computational overhead. The time and space required to train the generated model is no less than (or even greater than) that of training the ordinal regression model, yet the performance gain is not so significant.**
>
> We have rigorously evaluated the model's performance across a variety of benchmark datasets, consistently observing improvements in performance across all datasets and for every model variant tested. This highlights our significant contribution to the field of ordinal regression.
>
> Besides, our proposed data augmentation method stands out for its generality. Our approach can be broadly applied to distinguish ordinal content from other information types and to generate “correct” new examples accurately. While we primarily tested it on image data, the conceptual framework of our method is versatile enough to be adapted to non-image data as well. This adaptability offers advantages not commonly found in traditional data augmentation methods.
>
> Typically, existing augmentations in image data focus on altering factors like position, rotation, and color. However, these factors are specific to image data, and it remains unclear how to effectively augment non-image data. Our method transcends these limitations by automatically inferring latent ordinal content and other non-ordinal factors. By manipulating the inferred non-ordinal factors, we can generate new examples that maintain the crucial ordinal content. This capability makes our method broadly applicable and enhances the reliability of current machine learning techniques.
>
> ---
>
> **Q2. It is unclear whether the proposed framework could guarantee that to what extent the content information can be maintained in the invariant ordinal content factors z\_o\_tilde\_hat, which is crucial for the quality of generated images and further affects the performance of the ordinal regression model.**
>
> Our method is grounded in the theoretical framework of minimal change (Kong et al. 2022), which is instrumental in achieving disentanglement. According to the theoretical insights from prior research, our method's applicability to ordinal regression becomes evident. The principle implies that, given a sufficient number of images from different classes, the component $\hat{z}_o$ can be identified on a component-wise basis. This indicates that $\hat{z}_o$ undergoes a simple component-wise transformation solely from $z_o$. Consequently, $\hat{z}_o$ is devoid of information about the style component $z_n$, as it remains unaffected by $z_n$, thus achieving disentanglement.
>
> The following provides the intuitive explanation about disentanglement:
>
> - We categorize the latent factors into two groups: ordinal ($\hat{z}_o$) and non-ordinal ($\hat{z}_n$). In this framework, $\hat{z}_o$ serves as the repository for crucial ordinal content information that determines the ordinal category.
>
> - For reconstruction, we utilize both the predicted ordinal factors ($\hat{z}_o$) and the predicted non-ordinal factors ($\hat{z}_n$). $\hat{z}_o$ is specifically employed for predicting ordinal labels and is subjected to minimal change, restricting its influence in generating new instances.
>
> - Assuming the generative model attains minimal reconstruction error, $\hat{z}_o$, the factor designated for ordinal label prediction, effectively captures the ordinal content. By constraining it through the principle of minimal change, it encapsulates only the vital information pertinent to ordinal labels, while $\hat{z}_n$ accommodates other types of information.
>
> - This approach leads to a clear disentanglement of ordinal and non-ordinal information, enhancing the model's effectiveness.
>
> ---
>
> **References**
>
> Kong, L., Xie, S., Yao, W., Zheng, Y., Chen, G., Stojanov, P., Akinwande, V., & Zhang, K. (2022). Partial disentanglement for domain adaptation. In K. Chaudhuri, S. Jegelka, L. Song, C. Szepesvari, G. Niu, & S. Sabato (Eds.), Proceedings of the 39th International Conference on Machine Learning (Vol. 162, pp. 11455–11472). PMLR.

---

> ### Author Response · Authors · 2023-11-19
> **Responses from the authors - Part II**
>
> **Q3. Do the performance of the model sensitive to the setting of $\displaystyle \lambda _{1}$ in Eq. (4)?**
>
> We are running sensitivity analysis on $\lambda _1$ for analyzing the performance of OCP-CL. This section and the paper will be updated after we obtain the results.
>
> ---
>
> **Q4. The authors choose GAN as the generative model. It seems that the proposed data generative process could also be implemented by other model families (e.g., VAE or diffusion models).**
>
> Our proposed method can be seamlessly integrated with a variational autoencoder (VAE). Specifically, the mask can be applied directly to the latent factors $\hat{z}$ derived from the encoder.
>
> Exploring the application of our method in a diffusion model context is intriguing due to its robust capabilities in image generation. However, our method's direct implementation faces challenges in this setting. The denoising process in diffusion models may lead to inconsistent feature disentanglement, where the ordinal content dimensions in the latent variables do not align consistently across different denoising steps. This issue necessitates additional research to develop consistency measures that would enable the effective application of our method to diffusion models.

---

> ### Author Response · Authors · 2023-11-22
> **Awating for Your Valuable Feedback**
>
> Dear Reviewer PxYT,
>
> We appreciate the valuable comments and insights you have provided on our paper. In response to your feedback, we have made efforts to address all of your concerns comprehensively.
>
> The rolling discussion period will be close soon, we would like to hear your feedback.
>
> Could you please let us know if there are any questions or aspects of our paper requiring further clarification. We are more than willing to provide the necessary information and detail.
>
> Warm regards,
>
> The Authors

---

> ### Comment · Reviewer_PxYT · 2023-11-22
> **Response to the Authors**
>
> Thank the authors for their detail replies. My concerns are well addressed. Currently, I raise my score to 6.

---

> > ### Author Response · Authors · 2023-11-23
> > **Thank you for your Valuable Comments**
> >
> > Dear Reviewer PxYT,
> >
> > We are glad that all your concerns have been properly addressed. We wish to express our sincere gratitude for your effort of reviewing our paper and for providing such valuable feedback.
> >
> > Warm regards,
> >
> > The Authors

---

### Official Review · Reviewer_gPMn · 2023-11-07

**Soundness:** 3 good
**Presentation:** 3 good
**Contribution:** 3 good
**Rating:** 6
**Confidence:** 4

**Summary:**

The motivation of this paper is very clear and has strong guiding significance for applying contrastive learning to the related field of Ordinal Regression. In view of the fact that strong data enhancement methods in existing contrastive learning will destroy or weaken localized and subtle ordinal content information, this paper proposes a generative data augmentation method that decouples ordinal content factors and non-ordinal content factors. In this method, the author adopts the principle of minimum change for variables related to ordinal content to maintain the invariance of ordinal content during the generation process. A series of experiments demonstrate the effectiveness of the proposed generative data augmentation approach.

**Strengths:**

1. The motivation of the paper is clear and the generative data augmentation method that decouples ordinal content factors and non-ordinal content factors is quite novel. I believe that the generative data augmentation method proposed in this article is more advanced than traditional methods such as Gaussian blur and color dithering, which will provide a new sight for both ordinal regression and contrastive learning community.
2. The manuscript is well organized and thus it is clear and easy to understand.
3. The experiments in this paper are sufficient, which fully demonstrates the effectiveness of the model.

**Weaknesses:**

1. The symbols in Section 3.1 are very confusing. Some mathematical symbols add hat, some add tilde, and some add both at the same time, which is easy to confuse readers.
2. As we all know, it is difficult to adjust the parameters of generative adversarial models in most practical applications. Therefore, I hope the authors can tell me how difficult it is to tune the parameters of the proposed generative model, which is important for practical tasks.
3. I think the experiments in this article did not fully verify that zo is an ordinal content factor and zn is a non-ordinal content factor. For example, the former obtains wrinkles or gray hair through the generator alone, while the latter obtains other features unrelated to age. From my point of view, the experimental part can only prove that the generated data-augmented images can maintain the ordinal content, which may benefit from the powerful generation ability of the generator itself. So I'd like to see more visual experiments verify this.
4. In the data augmentation results produced in Figure 4, in the first group of augmentations from 4 to 6 years old, although the age range of the characters remains unchanged, the gender has changed. I think the ordinal invariant data augmentation produced by the proposed model may introduce additional noise, but there is no analysis of this additional noise in the paper. Will this additional noise generally affect the results?

**Questions:**

1. The symbols in Section 3.1 are very confusing. Some mathematical symbols add hat, some add tilde, and some add both at the same time, which is easy to confuse readers.
2. As we all know, it is difficult to adjust the parameters of generative adversarial models in most practical applications. Therefore, I hope the authors can tell me how difficult it is to tune the parameters of the proposed generative model, which is important for practical tasks.
3. I think the experiments in this article did not fully verify that zo is an ordinal content factor and zn is a non-ordinal content factor. For example, the former obtains wrinkles or gray hair through the generator alone, while the latter obtains other features unrelated to age. From my point of view, the experimental part can only prove that the generated data-augmented images can maintain the ordinal content, which may benefit from the powerful generation ability of the generator itself. So I'd like to see more visual experiments verify this.
4. In the data augmentation results produced in Figure 4, in the first group of augmentations from 4 to 6 years old, although the age range of the characters remains unchanged, the gender has changed. I think the ordinal invariant data augmentation produced by the proposed model may introduce additional noise, but there is no analysis of this additional noise in the paper. Will this additional noise generally affect the results?

---

> ### Author Response · Authors · 2023-11-19
> **Responses from the authors**
>
> **We thank the reviewer for providing these valuable comments.**
>
> Below, we provide detailed responses to address each of the concerns raised.
>
> ---
>
> **Q1. The symbols in Section 3.1 are very confusing. Some mathematical symbols add hat, some add tilde, and some add both at the same time, which is easy to confuse readers.**
>
> Thank you for highlighting the potential confusion in our notation. To resolve this issue, we will revise the notation in our paper: the invariant ordinal content factors $\tilde{z}_o$ will be updated to $z_v$, and $\hat{\tilde{z}}_o$ to $\hat{z}_v$. This change is intended to clearly differentiate them from $z_o$ and $\hat{z}_o$. We hope it will help in reducing confusion and improving the clarity of our notation.
>
> ---
>
> **Q2. As we all know, it is difficult to adjust the parameters of generative adversarial models in most practical applications. Therefore, I hope the authors can tell me how difficult it is to tune the parameters of the proposed generative model, which is important for practical tasks.**
>
> Training GANs has become more stable with the advent of StyleGAN models. However, hyperparameter tuning remains crucial, as improper settings can lead to failure of convergence. Our strategy involves using hyperparameters from similar, GAN-friendly datasets as a starting point.
>
> For instance, in the context of the age estimation dataset, we draw inspiration from the "CelebA" image generative dataset, renowned for its application in GANs. Both "CelebA" and our dataset feature similar attributes, notably the facial images of humans and celebrities. Therefore, the established hyperparameters for "CelebA" serve as an excellent baseline for training the "Adience" age estimation dataset.
>
> ---
>
> **Q3. I think the experiments in this article did not fully verify that zo is an ordinal content factor and zn is a non-ordinal factor. For example, the former obtains wrinkles or gray hair through the generator alone, while the latter obtains other features unrelated to age. From my point of view, the experimental part can only prove that the generated data-augmented images can maintain the ordinal content, which may benefit from the powerful generation ability of the generator itself. So I'd like to see more visual experiments verify this.**
>
> In the updated supplementary section, we present additional visual experiments that involve combining the $\hat{z}_o$ of various age groups with a constant $\hat{z}_n$. These experiments reveal that the age of the same individual changes to match the corresponding age group of $\hat{z}_o$. This observation confirms that $\hat{z}_o$ represents an ordinal content factor, while $\hat{z}_n$ is a non-ordinal factor.
>
> ---
>
> **Q4. In the data augmentation results produced in Figure 4, in the first group of augmentations from 4 to 6 years old, although the age range of the characters remains unchanged, the gender has changed. I think the ordinal invariant data augmentation produced by the proposed model may introduce additional noise, but there is no analysis of this additional noise in the paper. Will this additional noise generally affect the results?**
>
> We categorize the latent factors into two categories: (1) ordinal content factors, which are the latent factors related to ordinal classes, and (2) non-ordinal factors, which encompass the remaining factors containing only styling information.
>
> We would like to clarify that in the context of age estimation, gender is not considered an ordinal content factor. Introducing variability in non-ordinal factors, such as gender, can actually enhance the performance of the ordinal regression model. This is because it helps prevent the model from relying on non-ordinal factors for predictions. Consider a simplistic scenario where all training samples in the 4-6 age group are boys. In such a case, the model might incorrectly use gender as a predictor for this age group. Our method, by introducing diversity without altering the data's semantics, provides a more robust strategy for training ordinal regression models.

---

> > ### Author Response · Authors · 2023-11-23
> > **Thank you for your Valuable Comments**
> >
> > Dear Reviewer gPMn,
> >
> > We hope that all your concerns have been properly addressed. We wish to express our sincere gratitude for your effort of reviewing our paper and for providing such valuable feedback.
> >
> > Warm regards,
> >
> > The Authors

---

### Official Review · Reviewer_SPqB · 2023-11-09

**Soundness:** 2 fair
**Presentation:** 4 excellent
**Contribution:** 2 fair
**Rating:** 6
**Confidence:** 4

**Summary:**

The authors aim to enhance contrastive learning methods for ordinal regression tasks. They propose to disentangle ordinal content from non-ordinal content in latent factors and focus on augmenting non-ordinal information. Experiments on 3 public datasets are conducted to demonstrate the effectiveness of the proposed method.

**Strengths:**

1. The proposed method can be easily integrated into existing methods.

2. The experiment results look promising.

**Weaknesses:**

**Majors:**

1. If one latent feature (non-ordinal content) does not contribute much to an ordinal regression downstream task, then how much help could its augmentation provide? I would like to see some analysis about it.

2. The authors use an example in Figure 1 to show that "the commonly used strong augmentations can distort or even erase these essential features in ordinal regression data." Can you try some strong augmentation methods to demonstrate this claim in Tables 1, 2, and 3?

3. What about the performance of OCP-CL when the mask sparsity ($\lambda_1$) changes?

4. For results in Tables 1-4 and Figure 6, are they average of multiple runs? What about the standard deviations?

**Minors:**

5. The proposed OCP-CL is somehow similar to feature selection for content disentangling. Can feature selection methods be applied to learn ordinal and non-ordinal content?

6. How did you get the numbers in Table 5? By setting a threshold for $M$?

**Questions:**

Please see the weaknesses part. I would be inclined to increase my rating if the questions in the weaknesses part were well addressed and explained.

---

> ### Author Response · Authors · 2023-11-19
> **Responses from the authors**
>
> **We thank the reviewer for providing these valuable comments.**
>
> Below, we provide detailed responses to address each of the concerns raised.
>
> ---
>
> **Q1. If one latent feature (non-ordinal factors) does not contribute much to an ordinal regression downstream task, then how much help could its augmentation provide?**
>
> We categorize the latent factors into two categories: (1) ordinal content factors, which are the latent factors related to ordinal classes, and (2) non-ordinal factors, which encompass the remaining factors containing only styling information.
>
> Suppose that 'contribution' in the original question is defined as the proportion of a certain type of feature present in an image, e.g., features unrelated to its ordinal class occupy only a small portion of the image.
>
> In such cases, it is still challenging to guarantee that standard data augmentation will not distort ordinal information. This issue concerns not just proportion but also whether conflicts arise between certain augmentation methods and ordinal content. For example, in age estimation tasks, a person's hair color might be related to their age. This naturally conflicts with color distortion augmentation and is irrelevant to the proportion of 'hair' contributing to the image.
>
> ---
>
> **Q2. The authors use an example in Figure 1 to show that "the commonly used strong augmentations can distort or even erase these essential features in ordinal regression data." Can you try some strong augmentation methods to demonstrate this claim in Tables 1, 2, and 3?**
>
> Thank you for the suggestion. We will include visual analyses of all three datasets in the supplementary material to demonstrate how strong data augmentation might distort the ordinal content features visually.
>
> ---
>
> **Q3. What about the performance of OCP-CL when the mask sparsity ($\displaystyle \lambda _{1}$) changes?**
>
> The performance of OCP-CL varies with changes in mask sparsity. Table 5 summarizes the relationship between ordinal content and the number of non-ordinal dimensions when adjusting $\lambda_1$. If $\lambda_1$ is set too low, most latent dimensions are categorized as ordinal content factors, thereby failing to preserve the true ordinal content. Conversely, if $\lambda_1$ is too high, the diversity of augmentation is restricted, as only a small number of dimensions are identified as non-ordinal factors for a style change.
>
> We will provide a sensitivity analysis on $\lambda_1$ to analyze its impact on OCP-CL's performance in the paper.
>
> ---
>
> **Q4. For results in Tables 1-4 and Figure 6, are they average of multiple runs? What about the standard deviations?**
>
> For the age estimation task, we employ a five-fold cross-validation protocol in line with previous works (Li et al., 2021; Shen et al., 2018). The results reported are the average of the five folds. For the diabetic retinopathy rating and weather condition prediction tasks, the results are presented as averages of five runs. The standard deviations will be included in the updated version of the paper.
>
> ---
>
> **Q5. The proposed OCP-CL is somehow similar to feature selection for content disentangling. Can feature selection methods be applied to learn ordinal and non-ordinal factors?**
>
> Traditional feature selection methods are applicable when features are predefined. These methods can identify the most relevant features for ordinal regression by leveraging various dependence relations within the data. However, in scenarios where features are not explicitly provided, such as with raw image data, selecting features on a pixel-by-pixel basis is impractical. Instead, it is more effective to first extract latent features (or concepts) before proceeding with feature selection.
>
> Our method can be regarded as a feature selection approach tailored for latent features. It is specifically designed to automatically disentangle ordinal content factors (features) and non-ordinal factors based on the principle of minimum change. This strategy ensures that the identified ordinal content factors are pertinent and relevant for ordinal regression.
>
> ---
>
> **Q6. How did you get the numbers in Table 5? By setting a threshold for $M$?**
>
> We adjust the hyperparameter $\lambda_1$ to control the sparsity of the mask. Table 5 presents the number of ordinal content dimensions in the latent space corresponding to different values of $\lambda_1$.
>
> ---
>
> **References**
>
> Li, W., Huang, X., Lu, J., Feng, J., & Zhou, J. (2021). Learning probabilistic ordinal embeddings for uncertainty-aware regression. Proceedings of the IEEE/CVF Conference on Computer Vision and Pattern Recognition, 13896–13905.
>
> Shen, W., Guo, Y., Wang, Y., Zhao, K., Wang, B., & Yuille, A. L. (2018). Deep regression forests for age estimation. Proceedings of the IEEE Conference on Computer Vision and Pattern Recognition, 2304–2313.

---

> > ### Comment · Reviewer_SPqB · 2023-11-21
> > **Response to the Rebuttal**
> >
> > The authors' response addresses most of my concerns. However, Q1 is still my major concern. I believe the authors mistaken my question, so I clarify it as follows.
> >
> > The authors categorize the latent factors into ordinal and non-ordinal ones. In my understanding, the authors focus on augmenting the non-ordinal latent factors. As non-ordinal latent factors are not related to downstream ordinal tasks, why could the augmentations help improve ordinal classification/regression?

---

> ### Author Response · Authors · 2023-11-21
> **Response to Reviewer SPqB**
>
> **Thank you for the clarification and we are glad that our responses have addressed most of your concerns.**
>
> ---
>
> **Follow-Up Q1. The authors categorize the latent factors into ordinal and non-ordinal ones. In my understanding, the authors focus on augmenting the non-ordinal latent factors. As non-ordinal latent factors are not related to downstream ordinal tasks, why could the augmentations help improve ordinal classification/regression?**
>
>
> In this paper, we apply contrastive learning (Chen et al., 2020; He et al., 2020; Khosla et al., 2020) to the ordinal regression tasks, as it is a powerful training strategy that has demonstrated significant success in various vision applications. Within the contrastive learning framework, a strong data augmentation module is employed to create diverse positive and negative pairs for the learning objective. The module has been recognized as a beneficial component in the contrastive learning framework for conventional image data (Chen et al., 2020; Khosla et al., 2020). However, in the case of ordinal regression data, we found that strong augmentations can distort the ordinal content in the images, as illustrated in Figure 1. Our experiments, as shown in Table 4, reveal that inappropriate data augmentation can lead to a heavy degradation in performance for a contrastive learning framework. Therefore, we propose a safe data augmentation method that replaces the original strong augmentations in contrastive learning, enabling the ordinal regression framework to fully harness the power of contrastive learning. The results in Tables 1, 2, and 3 demonstrate the enhanced performance achieved by adapting contrastive learning objectives with our proposed generative data augmentation method.
>
> ---
>
> **We hope this has resolved your remaining concerns. Please feel free to let us know if you have any other questions.**

---

> ### Author Response · Authors · 2023-11-21
> **Response to Follow-Up Q1 (Continue)**
>
> **We think the aforementioned question is highly meaningful and we would like to provide some additional intuitions into why augmenting non-ordinal factors can enhance ordinal classification/regression.**
>
> *Generally speaking, data augmentation aims to modify the styling factors in original examples that are not related to the predictive objectives of downstream tasks* (Von Kügelgen et al., 2021).
>
> **In computer vision tasks**, by changing the styles in images, we add more variety to the training data. This helps the model not to focus too much on the specific styles it sees in the training images. It teaches the model to recognize objects or features in images, no matter how the style of the image changes. This is important because in the real world, images can come in many different styles. So, adding style changes in training helps the model perform well on all kinds of images (Shorten & Khoshgoftaar, 2019).
>
>
> **In the context of ordinal regression**, style information is referred to as non-ordinal information, governed by underlying non-ordinal factors. Our method also aims to enrich style diversity by altering non-ordinal information. This is achieved by randomly sampling non-ordinal factors while maintaining the ordinal content factors, then generating examples based on these factors. By preserving the ordinal content factors, we can change the image's style while keeping its ordinal content unchanged, thereby generating synthetic (counterfactual) images not seen in the training data. This approach enables neural networks to access more samples with diverse styles, thereby improving the generalization capabilities of ordinal regression models for unseen samples.
>
>
> As demonstrated in Figure 7, by randomly sampling non-ordinal factors while maintaining the ordinal content factors, we can alter various aspects of the image's style, such as people's dressing, background, and camera angles, etc., ensuring the ordinal content remains unchanged.
>
>
> It is also important to emphasize two major advantages of our data augmentation methods:
>
>
> - Firstly, existing image augmentation strategies do not guarantee the preservation of ordinal information during the augmentation process. For example, color jittering can change an image's color, potentially altering white hair to yellow, which could obscure the age of the person in the image.
>
>
> - Secondly, our proposed data augmentation method is general. Our approach can be broadly applied to automatically infer ordinal content from other types of information and generate new examples with guarantees. While primarily tested on image data, our method's framework should be adaptable to non-image data. This adaptability is not achievable with traditional data augmentation methods, which mainly focus on image data. For instance, applying rotations to non-image data is not feasible.
>
> ---
>
> **Reference**
>
> Von Kügelgen, J., Sharma, Y., Gresele, L., Brendel, W., Schölkopf, B., Besserve, M., & Locatello, F. (2021). Self-supervised learning with data augmentations provably isolates content from style. Advances in Neural Information Processing Systems, 34, 16451–16467.
>
> Shorten, C., & Khoshgoftaar, T. M. (2019). A survey on image data augmentation for deep learning. Journal of Big Data, 6(1), 1–48.

---

> > ### Comment · Reviewer_SPqB · 2023-11-21
> > **Response to the Rebuttal**
> >
> > Thanks for the detailed explanation. I upgraded my rating from weak reject to weak accept.

---

> > > ### Author Response · Authors · 2023-11-23
> > > **Thank you for your Valuable Comments**
> > >
> > > Dear Reviewer SPqB,
> > >
> > > We are glad that all your concerns have been properly addressed. We wish to express our sincere gratitude for your effort of reviewing our paper and for providing such valuable feedback.
> > >
> > > Warm regards,
> > >
> > > The Authors

---

### Meta-Review · Area_Chair_YvUQ · 2023-12-04

**Metareview:**

I have read all the materials of this paper including the manuscript, appendix, comments, and response. Based on collected information from all reviewers and my personal judgment, I can make the recommendation on this paper, *accept*. No objection from reviewers who participated in the internal discussion was raised against the accept recommendation.

**Research Question**

The authors consider the ordinal regression problem from the perspective of contrastive learning.

**Motivation**

The authors mentioned that strong augmentations in contrastive learning can easily overshadow or diminish this ordinal content information. I am convinced by the age estimation example in the introduction part.

**Philosophy**

The authors treat the features into two categories, ordinal information and non-ordinal information, which can be separated through the minimal change principle. The authors aim to preserve the ordinal information during the augmentation. The logic makes sense to me.

**Techniques**

The authors first analyzed the data generation process shown in Figure 2, which is good. Beyond the ordinal information and non-ordinal information, the authors also considered $z_v$, a comprehensive collection of ordinal -specific attribute, which is good. It is also nice to see Section 3.1 talks about the minimal change principle in the disentanglement process. Based on that, the proposed technique is in general sound to me.

**Experiments**

1. Significant tests in statistics are needed for Table 1-3.

2. Figure 4 is good to verify the motivation. However, it can be more stronger by some quantitively evaluations.

3. Some results in Appendix are more interesting to me, such as Figure 8 and 9. I suggest the authors to condense the descriptions and figures in the main paper and put these two figures back.

4. There are several additional experiments during the author-reviewer discussion period. Please add them into the Appendix as well.

**Presentation**

I have some minor comments on the presentations.

1. Figure 1 and 3 can be more dense.

2. Table 1, 2, and 3 can be merged into one big table.

3. Figure 4 should be annotated into three groups with texts.

4. "Fig. 2" should be "Figure 2."

5. Consider to use $\lambda$ instead of  $\lambda_1$ in Eq. (4).

6. The content in Appendix should be cited in the main paper. The first citation of appendix in the main paper is Appendix D in the beginning of Section 4.

**Justification For Why Not Higher Score:**

N/A

**Justification For Why Not Lower Score:**

This paper is self-standing, with a clear motivation, reasonable philosophy, sound technique, and enough evaluation, which I believe meets the bar of ICLR.

---

### Decision · Program_Chairs · 2024-01-16

Accept (poster)